



# Wildfire-atmosphere interactions during the Santa Coloma de Queralt fire: the development of a fire-induced circulation

Tristan Roelofs[1], Marc Castellnou[1,2], Jordi Vilà-Guerau de Arellano[1], Martin Janssens[1], and Chiel van Heerwaarden[1]

[1]Meteorology and Air Quality Group, Wageningen University & Research, The Netherlands
[2]Catalan Fire & Rescue Service, Bombers GRAF, Barcelona, Spain

**Correspondence:** Tristan Roelofs (Tristan.roelofs@wur.nl)

**Abstract.** High fireline intensities during extreme wildfire events can trigger pyro-convection, causing unpredictable fire spread behaviour, including faster-than-predicted fire spread and continued burning throughout the night. Earlier studies hypothesised that the main impact of pyro-convection on the fire behaviour is through the acceleration of the rear inflow. To assess this hypothesis, we used MicroHH to create a high-resolution (25 m) turbulence-resolving 3D large-eddy simulation (25.6 by 38.4 km$^2$) of the Santa Coloma de Queralt fire. We validated the in-plume virtual potential temperature using sounding measurements, to our knowledge, a novel approach for validating large-eddy simulations of pyro-convection. In-depth analysis of the wind patterns revealed an increase in rear inflow due to pyro-convection, as well as a frontal inflow of comparable magnitude, as part of a fire-induced circulation ahead of the fire. The frontal inflow could counteract the accelerated rear inflow and is associated with fire-generated vortices and long-range spotting. Additionally, we found that the fire-induced circulation simultaneously deepens and lowers the boundary layer in the 4 km ahead, thereby disrupting the transition from the convective daytime to a stably stratified nighttime boundary layer. This disruption provides a plausible explanation for the sustained nighttime burning during the Santa Coloma de Queralt fire. Therefore, we argue that the primary impact of pyro-convection on wildfire behaviour depends on the balance between wind patterns at the rear and in front of the fire (revised hypothesis), rather than solely on the acceleration of the rear inflow (original hypothesis).

## 1 Introduction

Extreme wildfire events occur when the fire line intensity of wildfires exceeds the extinguishing capacity of the fire service of 10 MW m$^{-1}$ (Tedim et al., 2018). During these events, the high fire line intensity can trigger significant upward convective motions (i.e. pyro-convection), which affect the wind speed and direction in the direct surroundings of the wildfire. The altered winds will change the fire behaviour (Sun et al., 2009), impacting the fire line intensity and, subsequently, feed back on the pyro-convection. This feedback loop is referred to as wildfire-atmosphere interactions and is associated with unpredictable wildfire behaviour. Recent extreme wildfire events in Spain (2021) and Portugal (2017) that triggered pyro-convection spread significantly faster than predicted (Commissão Téchnica Independente, 2017; Castellnou et al., 2022), with some unexpectedly continuing to burn throughout the night despite worsening burning conditions. A well-documented and measured example is



the first day of the Santa Coloma de Queralt (SCQ) fire in Catalonia on the 24th of July, 2021 (CFRS, 2022; Castellnou et al.,
2022). The fire spread up to 4 times faster than predicted and unexpectedly continued burning into the night, maintaining
a convective plume until midnight. The inability to extinguish these wildfires, combined with their unpredictability, results
in dangerous situations for firefighters. Hence, in this study, we aim to enhance our understanding of wildfire-atmosphere
interactions through pyro-convection, using the SCQ fire as a case study, as it is a rare instance of an extreme wildfire event
that has both well-documented fire behaviour and measured pyro-convection (Werth et al., 2016; Moisseeva and Stull, 2021).

Current operational fire behaviour models exclude the impact of pyro-convection (Kochanski et al., 2013) due to the poorly
understood underlying physical connections to fire behaviour, despite the recognition of its importance since the 1950s (Byram,
1954; Davis, 1959). Based on a review of observational literature on convective wildfires, Potter (2012a, b) proposes that the
upward mass flux due to pyro-convection increases the upwind surface inflow. It would explain the faster-than-predicted fire
spread since operational models do not account for the increased upwind inflow due to pyro-convection. This agrees with
recent in-plume sounding measurements of pyro-convection, suggesting a significant correlation ($R^2 = 0.9$) between the pyro-
convective strength and the mismatch between predicted and observed fire spread (Castellnou et al., 2022). The challenge
with observations is their limited spatial and temporal extent, which inhibits simultaneously measuring all wildfire-atmosphere
interactions during pyro-convection.

Turbulence-resolving Large-Eddy Simulation (LES) tools overcome this limitation by simulating pyro-convection in both
space and time. Realistic LES studies of past wildfire events show that wildfires can modify the surrounding wind patterns (e.g.
Peace et al., 2015, 2016; Filippi et al., 2018; Couto et al., 2024a; Roberts et al., 2024). The exact impact on the wind patterns
varies between cases due to the varying environmental conditions. This aligns with idealised LES studies, which show that the
interactions between wildfires and the local airflow depend on environmental factors such as the fuel type (Coen et al., 2013)
and atmospheric stability (Sun et al., 2009). Despite the variability, the LES studies agree with the theory proposed by Potter
(2012b) that wildfires can accelerate the upwind inflow (Coen et al., 2013; Peace et al., 2016; Filippi et al., 2018).

In addition to the accelerated upwind inflow, the simulations by Coen et al. (2013) and Peace et al. (2016) show that wildfires
can also modify the downwind airflow, creating frontal inflow, a feature of pyro-convection regularly used operationally to set
backing fires. This matches with Doppler measurements of convective wildfire plumes, which reveal significant frontal inflow
into the fire (Banta et al., 1992; Roberts et al., 2024). The measurements by Banta et al. (1992) suggest that the frontal inflow
is part of a downwind wildfire-induced circulation consisting of the convection inside the plume, downwind downdrafts and
the frontal inflow. The fire-induced circulation during convective wildfires could explain the continued nighttime burning, as
observed during the SCQ fire. At night, downdrafts transport relatively warm and dry air to the surface, which would counteract
the nighttime cooling and moistening of the atmosphere, thereby potentially mitigating the worsening burning conditions at
night. Unfortunately, neither Coen et al. (2013) nor Peace et al. (2016) investigated the ability of fires to modify downwind
burning conditions at night. Their simulations do not continue into the night, nor do they analyse the vertical airflow and
subsequent impact on the thermodynamic structure of the atmospheric boundary layer during the downwind reversal of the
surface winds.





Another remaining challenge with realistic LES studies of past wildfire events is their validation. The dangerous conditions during convective wildfires limit the number of measurements (Sun et al., 2006). Consequently, realistic LES are mainly validated using observed fire perimeters and/or ambient weather observations (e.g. Kochanski et al., 2013; Peace et al., 2015, 2016; Filippi et al., 2018; Campos et al., 2023). However, a correctly simulated ambient weather or fire perimeter does not necessarily ensure an accurate simulation of pyro-convection. A recent study by Roberts et al. (2024) demonstrated that even with a prescribed fire perimeter (e.g. Couto et al., 2024b), the simulated pyro-convection and subsequent impact on the surrounding wind patterns did not match Doppler measurements of the kinematic plume structure. Hence, explicitly validating simulated pyro-convection is critical to advancing our understanding of wildfire-atmosphere interactions.

In this study, we developed an observation-driven LES of the SCQ fire using in-plume sounding measurements provided by the Catalan Fire and Rescue Service (Castellnou et al., 2022). These measurements enabled us to validate the simulated pyro-convection by comparing the simulated and observed thermodynamic structure inside the fire-induced plume, which contrasts with Roberts et al. (2024), who focused on validating the kinematic plume structure. To our knowledge, soundings have not been previously used for the validation of LES of pyro-convection, despite being increasingly used in operations to aid decision-making (Ribau et al.). Therefore, the first objective of this study is to demonstrate the ability of LES to reproduce in-plume sounding observations of thermodynamic plume structure during extreme wildfire events. Using the validated simulation, we pursue a twofold objective: (1) to study the impact of pyro-convection on the surrounding wind patterns proposed by Potter (2012b) and found by previous LES studies (Coen et al., 2013; Peace et al., 2016; Filippi et al., 2018); and (2) to analyse the impact of the fire-driven wind patterns on the thermodynamic structure of the atmospheric boundary layer. To achieve these objectives, we utilised MicroHH to simulate the SCQ fire, as it is a proven turbulence-resolving LES tool for simulating convective and pollution plumes (van Heerwaarden et al., 2014; Ražnjević et al., 2022), which enables a three-dimensional analysis of wildfire-atmosphere interactions at the moment of in-plume measurements. Furthermore, MicroHH allows for a landscape-scale domain (order of 10 km) while supporting a high resolution (order of 10 m). Hence, we can simulate the large-scale ambient turbulent structures that define the atmospheric boundary layer while simultaneously resolving the small-scale turbulent structures inside the pyro-convective plume of the SCQ fire.

## 2 Methods

### 2.1 Santa Coloma de Queralt fire

The SCQ fire started on the 24th of July 2021 at 14 UTC (LT - 2). It spread eastward under a predominantly westerly wind and directly developed a convective plume (Fig. 1a). Occasional overshooting created short-lived pyrocumulus clouds, called overshooting pyrocumulus (oPyroCu), between 17 and 19 UTC. The Catalan Fire and Rescue Service reports the arrival of the sea breeze at 17 UTC, explaining the formation of the oPyroCu (CFRS, 2022). After 19 UTC, no further pyrocloud formation was observed, which coincides with the stabilisation of the atmosphere during the day-to-night transition. It is expected that the stabilisation of the atmosphere limited the vertical extent of the plume, inhibiting the plume from reaching the lifting







**Figure 1. (a)** The fire spread during the first day of the SCQ fire in UTC (LT - 2), including the locations of the sounding (star) and the synoptic weather station (hexagon). **(b)** The rising speed of the sounding that is used to determine whether a sounding is in or outside the convective plume based on a 2 m s$^{-1}$ threshold (red dotted line). **(c)** The diurnal evolution of the temperature and relative humidity measured by the synoptic weather station. The sounding measurements and synoptic observations are part of a larger observational dataset described by Castellnou et al. (2022). The fire perimeter map is an updated version of the map provided by Castellnou et al. (2022), including recent insights into the fire spread behaviour of the SCQ fire.



condensation level. Nevertheless, the SCQ fire maintained a dry convective plume until midnight (22 UTC), despite the cooling and moistening around sunset (19:41 UTC; Fig. 1c).

At 19:51 UTC, a sounding as released at the right flank of the fire (yellow star; (Fig. 1a) that measured vertical profiles of temperature, relative humidity and rising speed. The measured profiles begin 80 m above ground level (e.g. Fig. 1b). In ambient conditions, the maximum rising speed of a sounding is $\pm 2$ m s$^{-1}$, which means that the sounding measured inside the convective part of the plume between 0.2 and 1.4 km altitude (Fig. 1b). Besides measuring the plume during its ascent, a partial vertical profile of the environment was measured during its descent around 20:30 UTC (not shown). Consequently, we can validate both the simulated convective plume and the environment with the sounding measurements.

## 2.2 Simulation setup

This study focuses on correctly simulating and exploring the impact of pyro-convection on the atmosphere. Hence, in our simulation setup, we prioritise atmospheric realism while significantly simplifying the simulated fire behaviour to isolate the impact of the fire on the atmosphere.

To analyse the impact of the SCQ fire on the atmosphere, we performed two four-hour simulations of the 24th of July 2021 (16–20 UTC) with MicroHH (Heerwaarden et al., 2017): one without fire, *ref-run*, and one with fire, *fire-run*. The difference between the *ref-run* and the *fire-run* shows the impact of the SCQ fire. We use the last hour of the simulations (19–20 UTC) for the analysis as it matches the timing of the sounding measurement (19:51 UTC). The first two hours serve as spin-up time, after which the fire is initialised in *fire-run*. This provides an additional hour of spin-up time (18–19 UTC) to allow the convective plume to develop, ensuring a convective plume between 19 and 20 UTC.

To create as realistic meteorological conditions as possible in MicroHH, we use the ERA5 reanalysis data (Hersbach et al., 2020) as boundary conditions. ERA5 is considered one of the best currently available reanalysis datasets for studying convective environments (Taszarek et al., 2021). Additionally, the high vertical resolution (28 layers within the lower 2 km of the atmosphere) of ERA5 at an hourly interval means that it captures the pre-fire atmospheric boundary layer, which is the layer where the pyro-convection occurs. The ERA5 data is retrieved at the centre of the SCQ fire (latitude: 41.51775°, longitude: 1.494428°).

Figure 2 shows the ERA5 boundary conditions of the virtual potential temperature ($\theta_v$), specific humidity ($q$), wind speed ($U$) and wind direction ($U_{dir}$) at 16, 18 and 20 UTC. The $q$ and $U_{dir}$ show moistening of the CBL combined with backing winds, indicating a sea breeze in ERA5 after 16 UTC (Fig. 2b,d). Besides moistening, the sea breeze also causes cooling of the boundary layer, which combined with the shift from a positive to a negative surface heat flux between 19 and 20 UTC explains the transformation of the convective boundary layer at 16 UTC (blue line; Fig. 2a) into a neutral boundary layer ($0 < z < 0.5$ km) capped by a residual layer ($0.5 < z < 2$ km) between 19 and 20 UTC (purple dotted line; Fig. 2a), the period of interest for this case study of the SCQ fire. The reduction in turbulence is also visible in the vertical profile of $U$, which shows the development of a low-level jet at 20 UTC (purple dotted line; Fig. 2c).

The arrival of the sea breeze after 16 UTC aligns with the observations of the Catalan fire and rescue service described above. However, the backing of the wind in ERA5 is stronger than the observations by the local fire service, which indicate that even





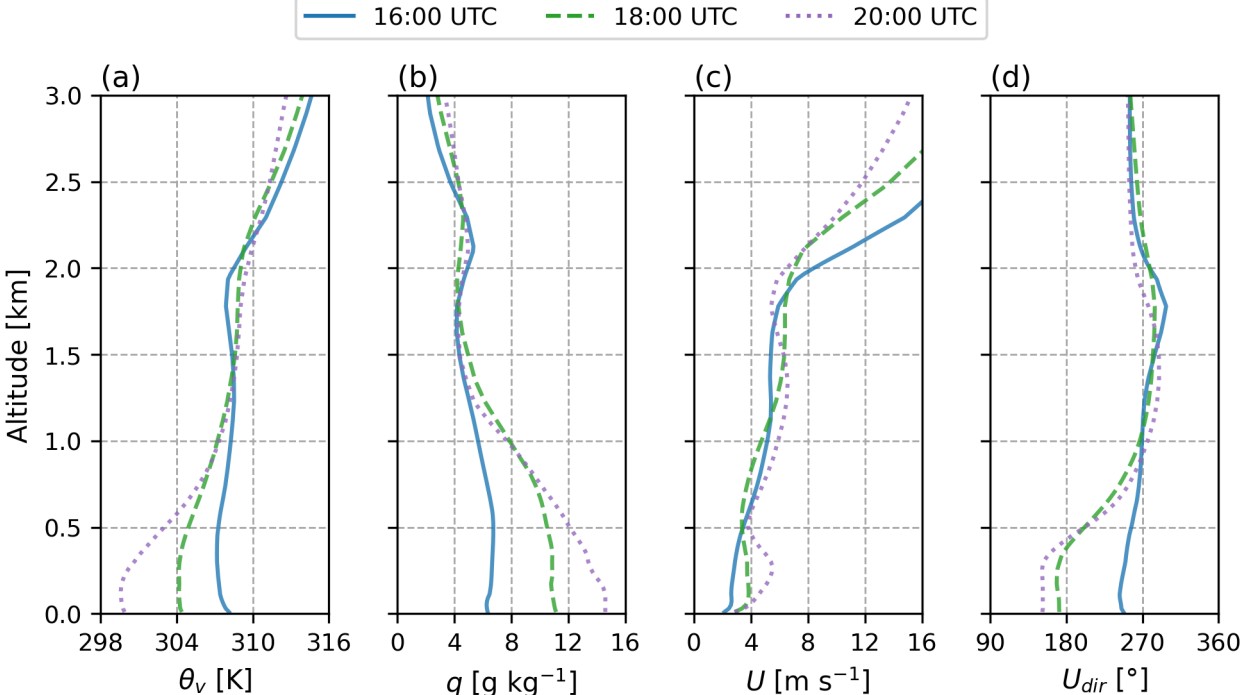

**Figure 2.** The boundary conditions for MicroHH of $\theta_v$ **(a)**, $q$ **(b)**, $U$ **(c)** and $U_{dir}$ **(d)** obtained from the ERA5 reanalysis dataset (Hersbach et al., 2020) at 16, 18 and 20 UTC.

after the sea breeze arrived, the fire continued to advance eastward under the influence of westerly winds. We expect that the

125 difference with the observations originates from the 30 km resolution of ERA5, which is too coarse to resolve the small-scale variations in wind direction due to the complex local topography. This is a common challenge with reanalysis products such as ERA5, but it does not affect the simulated fire behaviour, as we assume a stationary fire (detailed below). This allows us to focus on understanding the impact of pyro-convection on the atmosphere rather than accurately replicating the observed fire behaviour.

MicroHH uses periodic boundary conditions, meaning the simulated domain should be large enough to prevent unwanted recirculation of the wildfire-induced plume. Since the observations indicate a predominantly eastward development of the plume, we elongated the domain eastward. This resulted in a domain of 38.4 km × 25.64 km × 12 km (x × y × z; Fig. 3) with an equidistant horizontal resolution of 25m and a stretched vertical resolution starting at 10 m near the surface. Preliminary analysis showed that this domain size prevents unwanted recirculation of heat and moisture (see Appendix A).

To implement a fire within MicroHH, a surface area with increased sensible and latent heat flux is added. To represent the SCQ fire, we implemented a moon-shaped surface area measuring 1100m in width and 150 m in depth combined with a 145 kW m$^{-2}$ sensible heat flux (Fig. 3). The simulated fire also emits an inert tracer to visualise the simulated pyro-convection. To





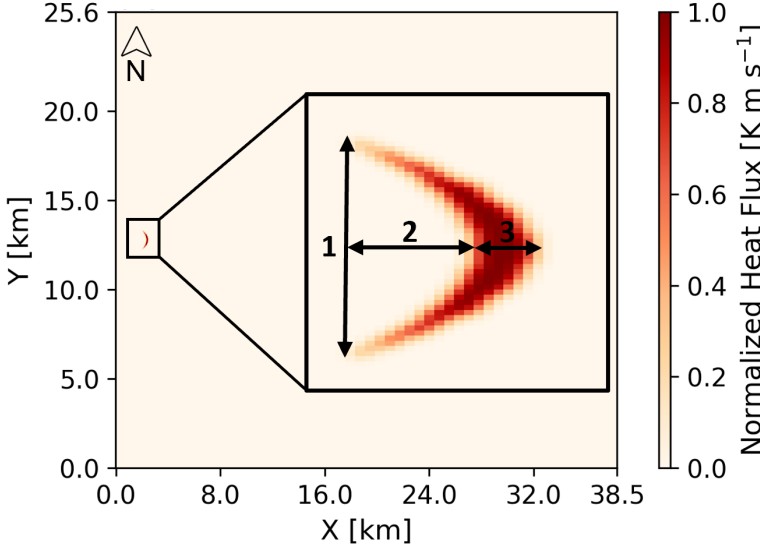

**Figure 3.** The normalised surface heat flux in the simulation using the maximum fire intensity (145 kW m$^{-2}$) with a zoom-in on the moon-shaped implementation of the fire. The arrows numbered one to three indicate the dimensions of the implemented wildfire, which are 1100 m, 350 m, and 150 m, respectively.

prevent numerical errors, we applied Gaussian smoothing ($\sigma$ = 25 m) to smooth the transition from the environmental fluxes ($\sim$ 1–10$^2$ W m$^{-2}$) to the wildfire fluxes ($\sim$ 10$^5$ W m$^{-2}$).

With this approach of implementing the SCQ fire into MicroHH, we made two significant simplifications: (1) we presume the fire to be stationary; (2) we use prescribed surface fluxes from the ERA5 reanalysis data surrounding the fire to negate the need for a radiation scheme. The stationarity is justified by the significantly smaller rate of spread of the fire compared to the wind speed, which makes the movement of the fire negligible from an atmospheric perspective. Furthermore, by using prescribed surface fluxes, we inherently neglect the plume shadowing effects in our simulations, which consequently removes

the need to simulate the aerosol composition of the plume. The lack of shadowing effects is expected to have a negligible impact on the outcomes of this study as we focus on the period between 19 - 20 UTC, which is around sunset (19:41 UTC). At sunset, the incoming radiation makes a minimal contribution to the energy balance, rendering the shadowing effect negligible. While these simplifications reduce the realism of certain aspects of the simulations (i.e. fire behaviour and radiation), they also reduce the complexity and uncertainty of the simulations, thereby facilitating the main objective of this study: assessing the

impact of pyro-convection on the surrounding atmosphere.





## 3 Results

The outcomes of this study are presented per objective. First, we compare the simulation with the visual observations and in-plume sounding measurements of the SCQ fire. Second, we analyse the impact of pyro-convection on the wind patterns and, subsequently, the thermodynamic structure of the atmospheric boundary layer surrounding the SCQ fire.

### 3.1 Validation

To validate the simulated pyro-convection between 19 and 20 UTC, we compared the simulated plume shape and vertical profile of the virtual potential temperature ($\theta_v$) with the observations in Fig. 4 and Fig. 5, respectively. The observed plume of the SCQ fire between 19 and 20 UTC is presented in Fig. 4a-c for 19:13, 19:34 and 19:59 UTC. Throughout the hour, a well-developed convection column was present (1) with occasional overshooting (2) and a horizontally dispersed smoke layer, also called the dispersion layer (3). Here, the distinction between the convection column and the dispersion layer is made visually based on the dominant axis of development: vertical (i.e., convection column) or horizontal (i.e. dispersion layer). The average simulated plume shape between 19 and 20 UTC based on the inert tracer is shown in Fig. 4d (grey outline). Similar to the observed plume, the average simulated plume also shows a convection column and a dispersion layer. Occasional overshooting also occurs in the simulation (not shown), but not consistently enough to affect the average plume shape. Additionally, the simulation shows, on average, downward transport of smoke below the dispersion layer, which suggests subsiding motions ahead of the fire front. Although the downward transport of smoke does not become apparent in Fig. 4a-c, it does match descriptions of the plume behaviour provided by local fire fighters.

To quantify the observed plume shape, we combined the measured rising speed and displacement of the in-plume sounding (black triangles; Fig. 4d). The rising speed indicated that the sounding was inside the convection column between 0.3 and 1.5 km, as its rising speed exceeded 2 m s$^{-1}$ (Fig. 1b). The displacement of the sounding (black triangles; Fig. 4d) between these altitudes shows that the simulated convection column up to 1.5 km is less tilted than observed. Above 1.5 km, the rising of the sounding effectively halts, resulting in approximately 3 km of primarily horizontal displacement between 1.5 and 1.7 km altitude before continuing its ascent (Fig. 4d). This suggests that the sounding exited the convection column between 1.5 and 1.7 km, but whether this was at the top of the plume cannot be derived from the rising speed and displacement of the sounding.

To further validate the simulated SCQ fire, we compared the simulated $\theta_v$ inside the plume and the environment with the measured $\theta_v$ during the ascent and descent of the sounding, respectively (black triangles; Fig. 5). The measurements near the surface (< 0.3 km) during the ascent show an increase in $\theta_v$ (Fig. 5b). This increase coincides with the near-surface acceleration of the rising speed (Fig. 1b), which indicates that it reflects the transition from the environment into the plume. Between 0.4 and 1.4 km, a relatively constant $\theta_v$ of 308 K is observed, which coincides with the observed convection column (i.e. where rising speed > 2 m s$^{-1}$). Above the well-mixed layer, an inversion is found (1.7 – 1.9 km) with a stable layer on top (> 1.9 km), which explains the predominantly horizontal displacement of the sounding for 3 km between 1.5 and 1.7 km altitude (Fig. 4d). The same inversion and the stable layer are also captured during the descent of the sounding (Fig. 5a). During the descent, the sounding measured the environmental conditions. Consequently, the measurements during the ascent above 1.5 - 1.7 km







**Figure 4.** A qualitative comparison between the observed and simulated plume shape between 19 and 20 UTC. The observed plume is presented before sunset (19:41 UTC) at 19:13 **(a)** and 19:34 **(b)** and after sunset at 19:59 **(c)** UTC (LT - 2) with the convection column, occasional overshooting and dispersion layer indicated by the numbers 1 to 3, respectively (Pictures taken from Ribau et al. (2022)). The average simulated plume shape between 19 and 20 UTC **(d)** is visualised using the inert tracer (> 0.0025) with the green and red shading indicating the convection column and dispersion layer. Additionally, the displacement of the sounding in the east-west direction is shown (black triangles).





represent the ambient conditions. Therefore, to validate the simulated plume, we compared the observed vertical profile of $\theta_v$

during the ascent between 0.4 and 1.5 km with the simulated $\theta_v$ inside the convection column.

A direct comparison of the simulation with the measured $\theta_v$ is impossible since the observed convection column was $\pm$ 30% more tilted than the simulated convection column (Fig. 4d). Consequently, the measurement locations (black triangles) do not align with the average location of the simulated convection column (green shaded area). To circumvent this mismatch, we compared the observations with the median $\theta_v$ profile (blue line) inside the average simulated convection column. The median

$\theta_v$ is determined per height inside the green-shaded area based on the averaged xz cross-section of $\theta_v$ at y = 12.8125 km. The spread in $\theta_v$ is indicated with the first and third quantiles (blue dotted lines). For consistency, the same approach is used to validate the simulated environment, except for using the *ref-run* instead of the *fire-run* and the median based on the entire xz cross-section instead of the green-shaded area.

Figure 5b shows the median profile of $\theta_v$ inside the average convection column up to 2.3 km, which is the average height

of the convection column between 19 and 20 UTC (Fig. 4d). At the surface, the median $\theta_v$ reaches 384 K (not shown), which

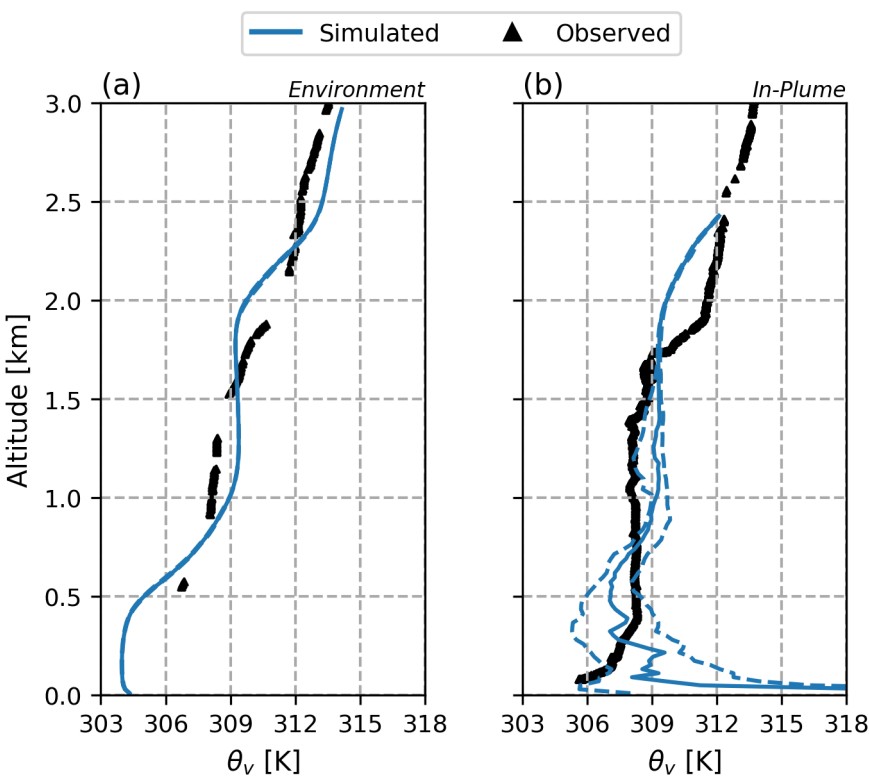

**Figure 5.** The vertical profiles $\theta_v$ measured during the descent (environment) and ascent (in-plume) of the sounding compared to the median simulated $\theta_v$ in the average environment **(a)** and the average convection column **(b)** between 19 and 20 UTC. The first and third quantiles (blue dashed lines) indicate the simulated variability.





quickly decreases due to mixing with the colder ambient air (304 K). This is opposite to the sounding observations (black triangles), which show a decrease in $\theta_v$ near the surface, because the simulated profile starts inside the flaming zone, whereas the sounding was launched outside at the flank of the flaming zone.

Above the rapid decrease in $\theta_v$, we find a well-mixed layer with a $\theta_v$ varying between 307 and 310 K in the simulation, which matches the observed mixed layer $\theta_v$ of 308 K (black triangles). However, the inversion capping the well-mixed layer is simulated $\pm\ 0.3$ km higher than observed (1.7 km). The same difference in inversion height is present in the environment (Fig. 5a).

The environmental conditions of the simulation are based on the ERA5 boundary conditions (Fig. 2), which also overestimate the inversion height by 0.3 to 0.4 km. Hence, the inversion height mismatch between the observations and the simulation (Fig. 5a) is created by ERA5 and does not reflect the performance of MicroHH. The overestimation by ERA5 is consistent with measurements in the same region and period of the SCQ fire presented by Mangan et al. (2023). They showed a consistent overestimation of the convective boundary layer by ERA5 due to the inability of ERA5 to capture the local surface heterogeneity. At night, this would result in an overestimation of the height of the residual layer, which we find in Fig. 5a.

Due to the overestimated inversion height, we conjecture that the simulation also overestimates the plume height. A higher plume could amplify the impacts of wildfire-induced pyro-convection on the surrounding kinematic and thermodynamic environment. Nonetheless, it is not expected to significantly alter the outcomes of this study, since we aim to explore the physical mechanisms behind the impacts of pyro-convection on the kinematic and thermodynamic structure of the atmosphere. In future studies, we will systematically quantify how, among others, the boundary layer structure alters the impact of pyro-convection on the atmosphere.

## 3.2 Fire-modified wind patterns

To investigate how the SCQ fire impacts its surrounding wind patterns, we compare the *fire-run* with the *ref-run*. The direct impact of any wildfire on the atmosphere is the creation of buoyancy by heating the air. For the SCQ fire, we find vertical velocities up to 3 m s$^{-1}$ at the surface (Fig. 6b), an order of magnitude larger than the environmental vertical velocities (Fig. 6a). To sustain the increased vertical airflow above the flaming zone (black dotted line), additional inflow into the flaming zone is needed. Figure 6b suggests two mechanisms that could provide the additional inflow: (1) downdrafts at the southern borders of the flaming zone and (2) horizontal inflow from the western, eastern and northern borders (grey streamlines).

To quantify the change in airflow, we calculate the mass balance of the red box surrounding the flaming zone (Fig. 6) between 0 and 60 m altitude, thereby focusing on the airflow changes in the near-surface layer below the plume neck (Fig. 7). Without the SCQ fire (*ref-run*), we find that the vertical in- and outflow are balanced ($149\times10^3$ and $-145\times10^3$ kg s$^{-1}$, respectively). Additionally, as expected, the horizontal part of the mass flux balance of the *ref-run* is dominated by the mean wind pattern that is the effect of a southerly sea breeze, with predominantly inflow from the south ($131\times10^3$ kg s$^{-1}$) and outflow in the north ($-132\times10^3$ kg s$^{-1}$).

Comparing the *ref-run* with the *fire-run*, we find a $620\times10^3$ kg s$^{-1}$ increase in vertical outflow, which is partially compensated by a $273\times10^3$ kg s$^{-1}$ increase in vertical inflow. The gap between the changes in vertical in- and outflow ($347\times10^3$ kg



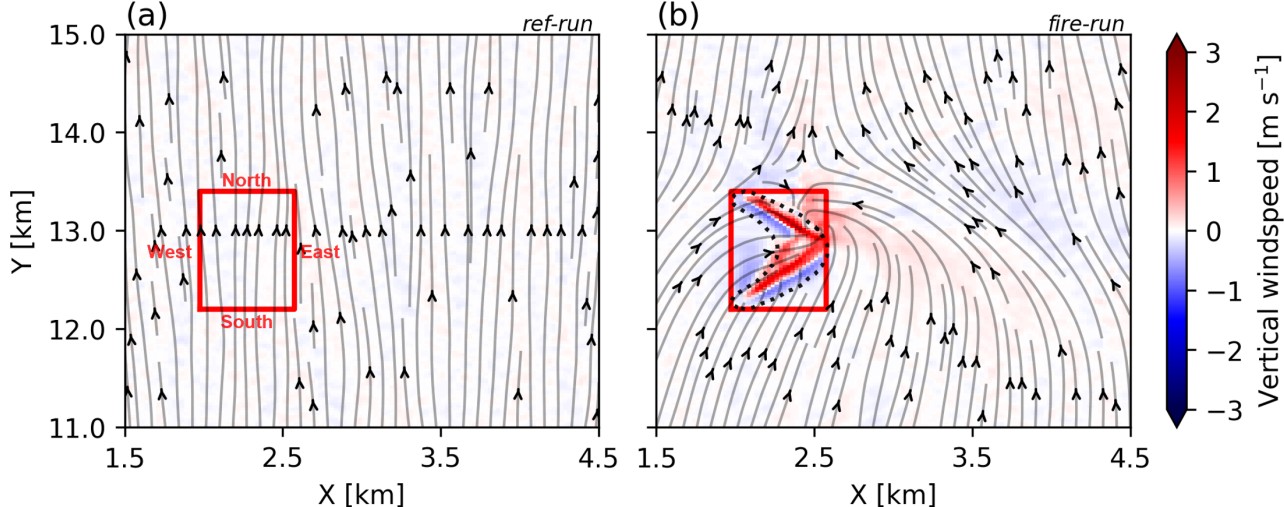

**Figure 6.** The average vertical wind speed in the *ref-run* **(a)** and *fire-run* **(b)** between 19 and 20 UTC at the lowest vertical level of the simulation (z = 10 m). The grey streamlines show the hourly-averaged horizontal airflow patterns, the magnitude of which is presented in Fig. 8a,c. Furthermore, the red rectangle is the area for which we calculated the mass balance to quantify the influence of the SCQ fire on the surface winds (Fig. 7), and the black dotted line **(b)** indicates the simulated flaming zone of the SCQ fire (Fig. 3).

s$^{-1}$) is covered predominantly by changes in the airflow over the northern and western borders (+129×10$^3$ and +176×10$^3$ kg s$^{-1}$, respectively). At the northern border, the SCQ fire increases the inflow while decreasing the outflow, changing it from a net outflow border (-132×10$^3$ kg s$^{-1}$) to an almost net zero border. At the western border, the fire predominantly increases the inflow (+145×10$^3$ kg s$^{-1}$). Consequently, the net vertical outflow induced by the SCQ fire (-343×10$^3$ kg s$^{-1}$) is mostly compensated by southern (172×10$^3$ kg s$^{-1}$) and western (168×10$^3$ kg s$^{-1}$) inflow.

The total mass flux balance for the *fire-run* does not indicate a significant contribution from the eastern border, the front of the SCQ fire, despite the streamlines suggesting significant changes in airflow at the eastern border due to the fire (Fig. 6). This contradiction arises because both the in- and outflow at the eastern border increase equally, effectively cancelling each other out in the total mass flux balance. To further analyse the airflow at the eastern border, we show the zonal and meridional wind separately in Fig. 8 for the *fire-run* (a & c). Additionally, we show the difference between the *fire-run* and the *ref-run* (b & d) to

highlight the acceleration and deceleration of the horizontal winds due to the SCQ fire. Figure 8b reveals two opposite changes at the eastern border: deceleration and acceleration of the zonal wind north and south of the centre of the fire (y = 12.8125 km; purple dash-dot line), respectively. This explains the simultaneous increase in in- and outflow at the east border.

   Although negligible for the net mass flux balance, the changes in the easterly in- and outflow affect the fire behaviour. Firstly, the additional inflow means an increase in oxygen supply for the fire, allowing for a higher combustion rate. In total,

the SCQ fire increased the inflow by more than 170%, meaning that the time in which the air inside the red box is refreshed decreases from 2.3 minutes (*ref-run*) to 0.8 minutes (*fire-run*). Secondly, spot fires, which were observed during the SCQ fire,




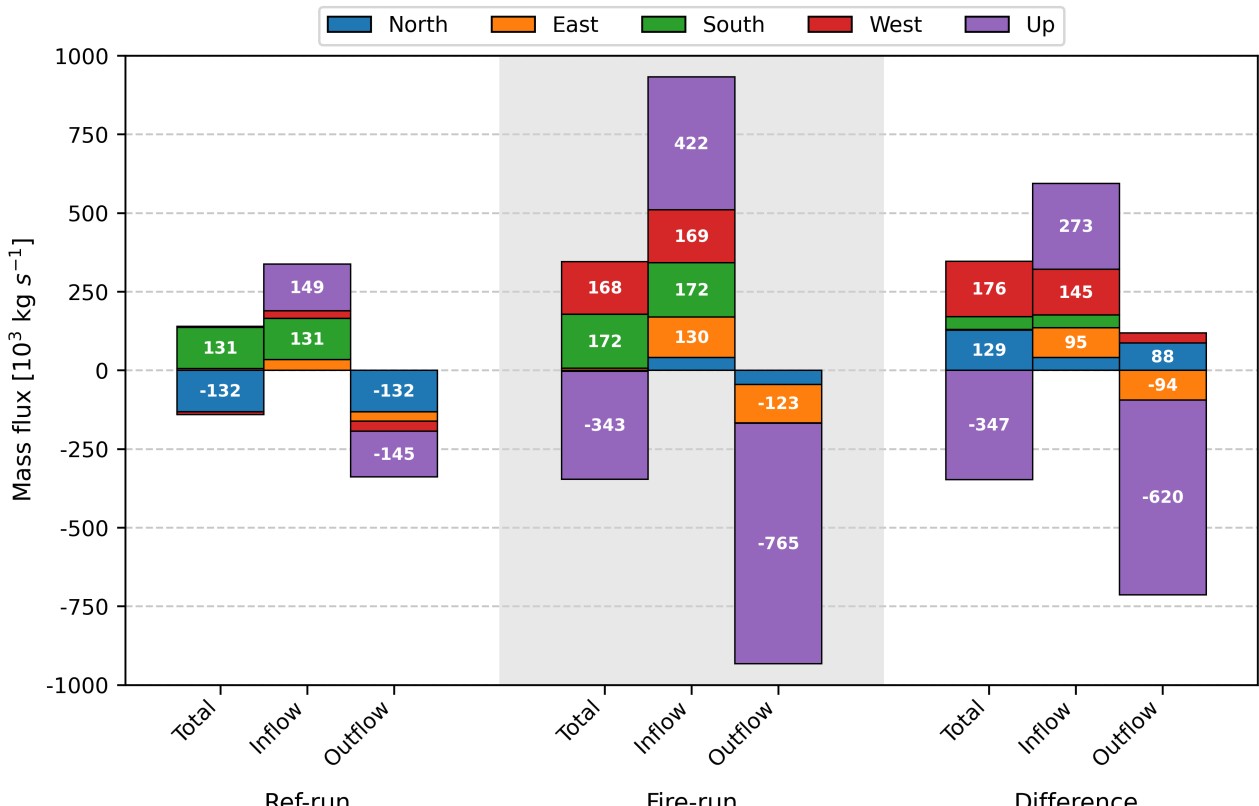

**Figure 7.** The mass flux balance for the *ref-run* and *fire-run* calculated over the boundaries of the red box in Fig. 6 in the lowest three layers of the simulation ($0 < z < 60$ m) between 19 and 20 UTC. The inflow (outflow) into the red box is defined as positive (negative). To investigate the impact of the SCQ fire on the mass transport, the difference between the two simulations (i.e., *fire-run - ref-run*) is shown.

will experience different wind directions depending on their location ahead of the main fire and therefore behave differently. To illustrate the relevance, we consider a simplified scenario in which the wind is the primary factor in determining the direction of the fire spread. In this scenario, a spot fire at the northern part of the front would spread in the opposite direction of the main fire (black dotted outline), thereby reducing fuel availability ahead. On the other hand, at the southern part of the front, the spot fires would move in the same direction, enhancing the rate of spread. Besides affecting the behaviour of spot fires, the changed winds ahead of the main fires also affect the way backing fires can be used to reduce the fuel availability.

Figure 9 shows the evolution of the surface wind patterns with altitude (Figs. 6 & 8). We limit ourselves to vertical cross-sections parallel to the fire spread direction (i.e., eastward) since the simulated plume also developed to the east. The southerly winds at the surface (Fig. 2d) would suggest otherwise, but the westerly winds above 500 m altitude (Fig. 2d) result in the plume developing towards the east, similar to the observations described in Sect. 2.1. Hence, the cross-sections parallel to the fire spread direction are the most relevant for further analysis.



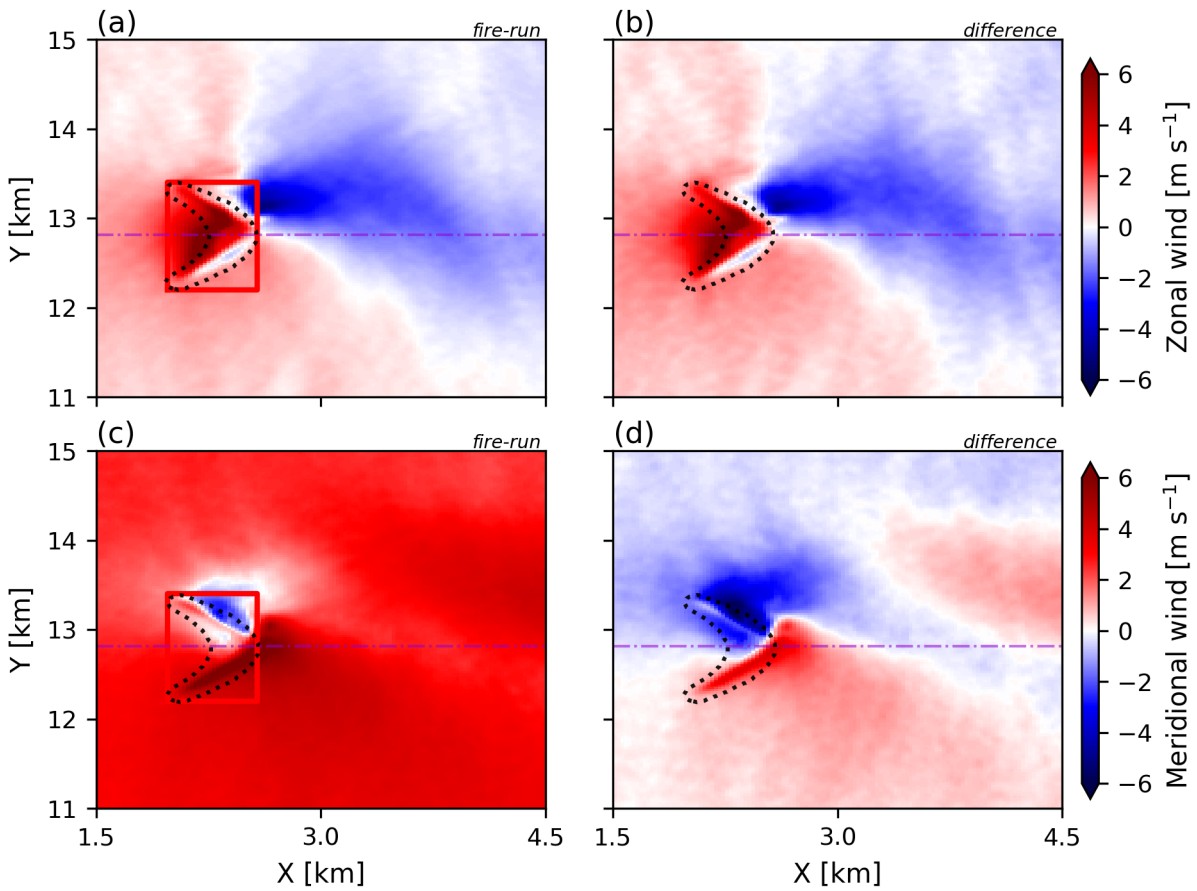

**Figure 8.** The average zonal **(a)** and meridional **(c)** wind between 19 and 20 UTC at the lowest vertical level of the simulation (z = 10 m). To visualise the acceleration and deceleration due to the SCQ fire, the average difference with the *ref-run* is shown for both the zonal **(b)** and meridional **(d)** wind. The black dotted line and the purple dash-dot line indicate the simulated flaming zone of the SCQ fire (Fig. 3) and the centre of the fire (i.e. y=12.8125 km). The red rectangle shows the boundaries over which the mass flux balance in Fig. 7 is calculated.



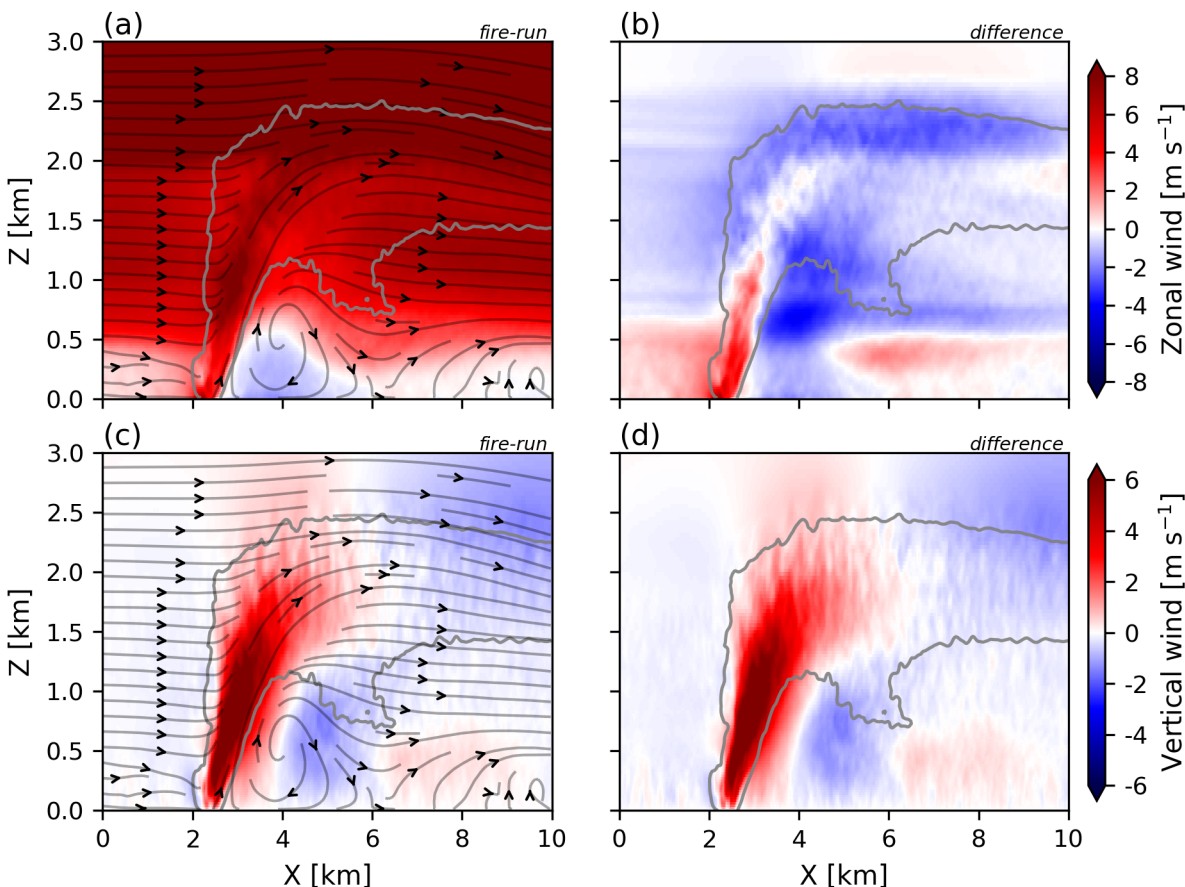

**Figure 9.** The vertical cross-sections of the average zonal **(a)** and vertical **(c)** wind between 19 and 20 UTC through the middle of the simulated fire (i.e. y = 12812.5 m). The grey outline and grey streamlines represent the average plume shape based on the inert tracer and the airflow through the cross-section. To visualise the acceleration and deceleration of the wind due to the SCQ fire, the average difference with the *ref-run* is shown for the zonal **(b)** and vertical **(d)** wind.





While both the acceleration and deceleration areas of the zonal wind extend up to 1.5 km altitude, their patterns differ (Fig. 9b). The acceleration occurs primarily inside the plume (grey outline), with the maximum acceleration at the surface. In contrast, the deceleration mainly occurs to the east of the fire (from x = 2.6 km onward), with the maximum deceleration between 0.5 and 1 km altitude.

The largest impact on the vertical wind is the acceleration inside the plume (Fig. 9d), resulting in instantaneous vertical wind speeds up to 32 m s$^{-1}$ (not shown). Additionally, Fig. 9d shows the development of downdrafts ahead of the fire between x = 4 and 6 km up to 1.2 km altitude.

The combined effect of these changes in zonal and vertical wind by the SCQ is visualised by the grey streamlines (Fig. 9a,c), which show the average airflow between 19 and 20 UTC. They reveal the formation of a fire-induced circulation east of the SCQ fire (x > 2.6 km) consisting of the rising motions inside the plume, downdrafts 2 km east of the fire and a negative zonal wind near the surface (z < 0.5 km). Furthermore, they show descending inflow west of the fire (x < 2 km) despite the relatively small negative vertical velocities upwind compared to the downdrafts ahead of the fire (Fig. 9c).

Considering that the SCQ fire spread eastward in reality (Fig. 1), these modifications of the airflow suggest that the SCQ fire also altered the thermodynamic structure of the atmospheric boundary layer ahead of itself. Changing the thermodynamic structure could potentially improve its future burning conditions (i.e., temperature, moisture, and stability), thereby explaining the continued burning of the SCQ fire until midnight.

## 3.3 Fire-modified boundary layer

To analyse the impact of the fire-modified wind patterns (Fig. 6 to 9) on the thermodynamic structure of the atmospheric boundary layer, we show the average vertical cross-sections of the virtual potential temperature ($\theta_v$) and specific humidity ($q$) during the *fire-run* in Fig. 10a,c. To highlight the impact of the SCQ fire on the thermodynamic structure, Fig. 10b,d shows the difference in $\theta_v$ and q between the *fire-run* and the *ref-run*.

Inside the plume (grey outline), we observe heating near the surface (<1 km; Fig. 10b), while moistening only begins above 0.5 km (Fig. 10d). This indicates that the fire directly affects $\theta_v$ through the sensible heat flux, while $q$ is primarily affected indirectly through the upward transport of ambient moisture by the fire-induced convection.

East of the plume (x > 2.6 km), two opposite patterns developed. Closest to the plume (2.6 < x < 4.5 km), Fig. 10b,d shows cooling and moistening in the lower part of the residual layer (0.5 < z < 1 km). Further eastward (4.5 < x < 8 km), the transition zone between the neutral boundary layer and the residual layer (0.25 < z < 0.75 km) is warmed and dried. These two opposite patterns reflect the interaction of the fire-induced circulation with the ambient boundary layer structure in which the SCQ fire occurred. Between 19 and 20 UTC, there is a neutral boundary layer (0 < z < 0.5 km) topped by a warmer and drier residual layer (0.5 < z < 2 km), which in turn is topped by an even warmer and drier free troposphere (Fig. 2a,b). Closest to the plume (2.6 < x < 4.5 km), the circulation transports relatively cool and moist air from the neutral boundary layer into the residual layer above. Further eastward (4 < x < 6 km), the downdrafts transport relatively warm and dry air downward from the residual layer to the neutral boundary layer. A similar pattern of warming and drying, albeit less strong, is present west of the plume due to the descending rear inflow.





**Figure 10.** The vertical cross-sections of the average virtual potential temperature, $\theta_v$ **(a)**, and specific humidity, $q$ **(c)**, between 19 and 20 UTC through the middle of the simulated fire (i.e. y = 12.8125 km). To highlight the impact of the SCQ fire, the average difference with the *ref-run* between 19 and 20 UTC is shown for $\theta_v$ **(b)** and $q$ **(d)**. The grey outline visualises the average shape of the simulated plume. Furthermore, the grey streamlines indicate the airflow during the simulated SCQ fire.



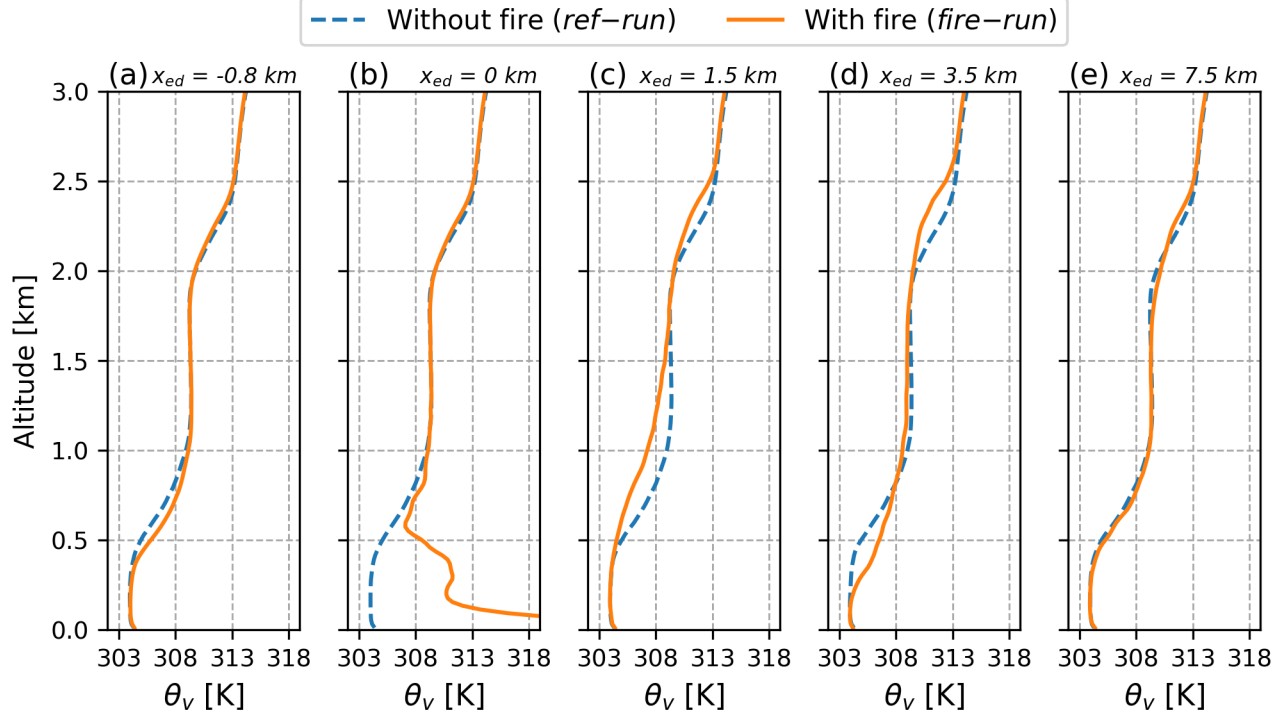

**Figure 11.** The average vertical profile of $\theta_v$ between 19 and 20 UTC for five eastward distances ($x_{ed}$) with respect to the head of the fire (x = 2.5 km, y = 12.8125 km): -0.8 km **(a)**, 0 km **(b)**, 1.5 km **(c)**, 3.5 km **(d)** and 7.5 km **(e)**. These five locations are visualised in Fig. 12a with red hexagons. To show the impact of the SCQ fire on that atmospheric boundary layer structure, the vertical profiles of $\theta_v$ for the simulation with (orange) and without (blue dotted) a fire are shown.

To explore how the upward (downward) transport of relatively cold (warm) air (Fig. 10) affects the atmospheric boundary layer structure, we show the evolution of the average vertical profile of $\theta_v$ between 19 and 20 UTC as a function of the eastward distance ($x_{ed}$) from the fire (Fig. 11). Directly above the fire ($x_{ed} = 0$), the heating by the fire deepens the boundary layer from

0.5 km to 2 km (Fig. 11b). West of the fire ($x_{ed}$ = -0.8 km), the descending inflow lowers the neutral boundary layer height (Fig. 11a). East of the fire, two distinctly different effects associated with the fire-induced circulation become apparent before the impact of the fire on the atmospheric boundary layer structure becomes negligible at $x_{ed}$ = 7.5 km (Fig. 11e). Near the fire ($x_{ed}$ = 1.5 km), upward transport of relatively cold air (Fig. 11) leads to a deepening of the neutral boundary layer (Fig. 11c). Further eastward ($x_{ed}$ = 3.5 km), the downdrafts (Fig. 6) lower the boundary layer height ($h$) from 0.5 to 0.25 km (Fig. 11d).

To investigate the spatial extent of these changes in $h$, we calculated $h$ based on the minimum $\frac{dq}{dz}$ between 0.1 and 1.6 km altitude (Fig. 12). This altitude range is chosen to exclude the surface layer (z < 0.1 km) and the capping inversion on top of the residual layer (2 < z < 2.5 km). We find a decrease in $h$ at the rear of the flaming zone, matching the descending rear inflow (Fig. 10). A similar decrease in $h$ is visible at the northern and southern flanks of the SCQ fire, suggesting that a similar



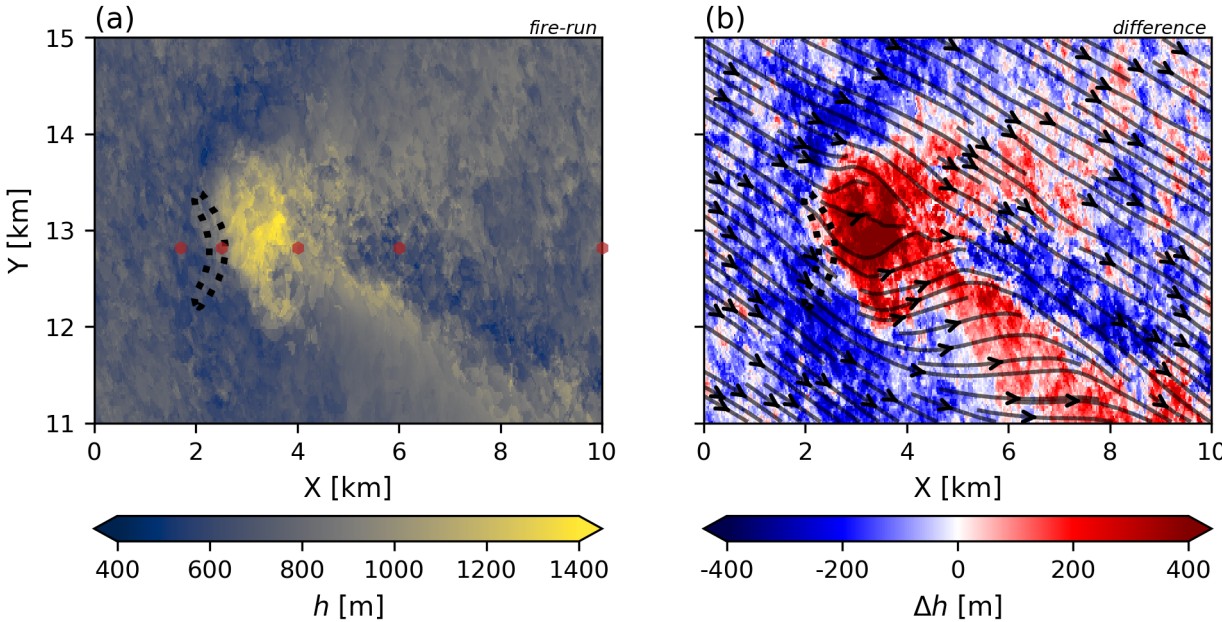

**Figure 12.** The average boundary layer height, $h$ **(a)** between 19 and 20 UTC based on the minimum $\frac{dq}{dz}$ between 0.1 and 1.6 km altitude. The red hexagons from left to right represent the locations of the vertical profiles **(a)** to **(e)** in Fig. 11. To highlight the impact of the SCQ fire on $h$, we calculated the difference in $h$ between the *fire-run* and *ref-run* **(b)**. The black streamlines represent the average airflow in the residual layer based on the simulated airflow at 950 m altitude.

descending inflow is present at the flanks. Ahead of the flaming zone (black dotted line), we find the same pattern as in Fig. 10
with an increase in $h$ within the first 2 to 3 km and subsequently a decrease in $h$. Figure 12b shows that both the increase
and decrease in $h$ are advected towards the southeast by the airflow above the convective boundary layer (black streamlines).
The southeast advection explains why we find no modification of the thermodynamic structure at 7.5 km ahead of the fire
(Fig. 11e). This lack of modification highlights the significance of directional wind shear in determining where and how far
ahead of itself a fire can alter the thermodynamic structure of the atmosphere. In case of the SCQ fire, the wind veered from
a southerly surface wind into a westerly to northwesterly wind above the boundary layer. Consequently, the change in $h$ was
advected towards the southeast, but with less (more) veering of the wind over height, the distance ahead of the SCQ fire with a
modified $h$ would be larger (smaller).

Besides affecting $h$, Fig. 11c & d shows that the SCQ fire also causes entrainment of free tropospheric air at the top of
the residual layer (2 < z < 2.5 km). Comparing the vertical profiles at $x_{ed}$ = 1.5 and 3.5 km reveals an increasing amount
of entrainment with increasing eastward distance from the fire. This is because the plume is tilted, which means that the
overshooting is tilted. Hence, the entrainment at the top of the residual layer is shifted in the direction of the wind, i.e. eastward
in the case of the SCQ fire.




## 4 Discussion

In this study, we demonstrated the use of in-plume sounding measurements to explicitly validate an LES of the pyro-convection
induced by the SCQ fire. Using the validated LES, we investigated the impact of pyro-convection on the wind patterns and
subsequently the thermodynamic structure of the atmospheric boundary layer surrounding the SCQ fire. In this chapter, we
reflect on our results and methodology in comparison to previous work and outline implications for future research.

### 4.1 Fire-modified wind patterns

Figure 13b summarises the key impacts of pyro-convection on the surrounding wind patterns found in our simulation of the
SCQ fire (Fig. 8 to 10). The convection inside the plume (1) created an accelerated descending rear inflow (2) and the formation
of a circulation ahead of the fire consisting of the rising plume (1), downdrafts two km ahead of the fire (3) and frontal inflow
(4).

The acceleration of the rear inflow in our simulation aligns with previous studies (Coen et al., 2013; Peace et al., 2016;
Filippi et al., 2018) and confirms the theory proposed by Potter (2012a, b) that the upward mass flux due to pyro-convection
accelerates the rear inflow (Fig. 13a). As discussed in the introduction, this theory provides a plausible explanation for the
faster-than-predicted fire spread observed for extreme wildfire events that trigger pyro-convection. Current operational fire
spread models do not account for the accelerated rear inflow due to pyro-convection (Kochanski et al., 2013). Hence, the

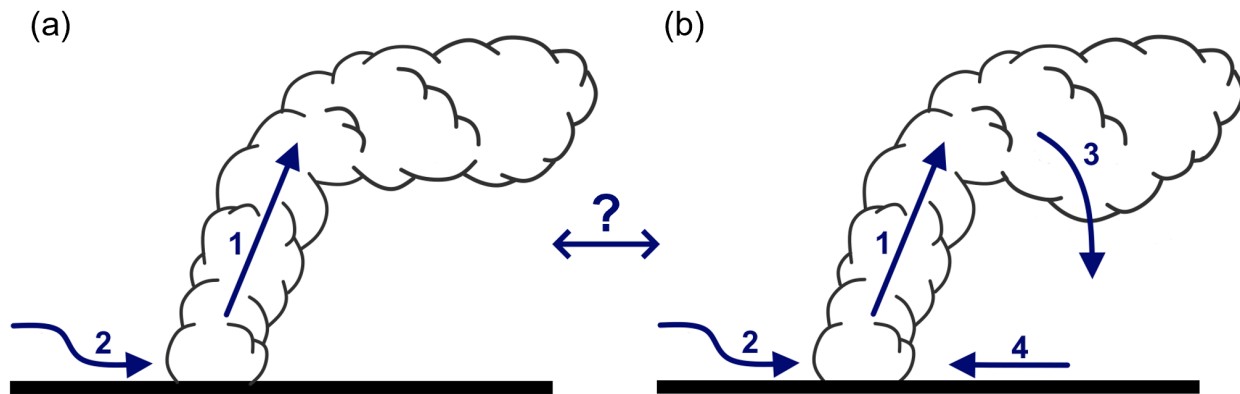

**Figure 13. (a)** A schematic overview of the impacts of pyro-convection on the wind patterns following Potter (2012a, b): (1) convective
motions inside the plume and (2) accelerated and descending rear inflow. **(b)** An extended version of **(a)** including the fire-induced circulation,
which consists of (1) convective motions inside the plume, (3) downdrafts ahead of the fire and (4) frontal inflow. The extended schematic is
based upon the outcomes of the SCQ fire simulations (Fig. 6) combined with previous observations (Banta et al., 1992; Roberts et al., 2024)
and simulations (Sun et al., 2009; Coen et al., 2013; Peace et al., 2016). The question mark highlights the unknown factors that govern the
development of fire-induced circulations, as they do not appear in all previous studies (e.g. Filippi et al., 2018).





wind experienced by the fire is underestimated by these models, which results in an underestimation of the fire spread. Recent in-plume measurements (Castellnou et al., 2022) confirmed this theory by showing a positive correlation between the
pyro-convective strength and the mismatch between predicted and observed fire spread; The stronger the pyro-convection, the stronger the acceleration of the rear inflow, which means a larger underestimation of the fire spread by the operational models (i.e. larger mismatch between predictions and observations).

However, our results suggest that this explanation for the mismatch between predicted and observed fire behaviour is incomplete. In addition to the acceleration of the rear inflow described by Potter (Fig. 13a), we find the development of a fire-induced
circulation ahead of the fire (Fig. 13b). The existence of such a fire-induced circulation has been (in)directly suggested in previous studies. Based on Doppler measurements of the horizontal wind, Banta et al. (1992) suggested the presence of a circulation ahead of the fire. Furthermore, both observations (Roberts et al., 2024) and simulations (Sun et al., 2009; Coen et al., 2013; Peace et al., 2016) show the development of frontal inflow, suggesting the presence of a fire-induced circulation. Coen et al. (2013) suggested that the frontal inflow can alter the fire behaviour by countering the acceleration of the rear inflow.
Additionally, radar observations indicate that frontal inflow is linked to the formation of fire-generated vortices (Lareau et al., 2022). This matches operational observations, which suggest that frontal inflow is typically associated with the occurrence of fire-generated vortices and long-range spot fires, two phenomena that were also observed during the SCQ fire. Therefore, based on our findings combined with the previous observations (Banta et al., 1992; Lareau et al., 2022; Roberts et al., 2024), simulations (Sun et al., 2009; Coen et al., 2013; Peace et al., 2016) and operational observations, we propose to extend the
theory provided by Potter (2012a, b) to include the fire-induced circulation (Fig. 13) to explain the mismatch between observed and predicted fire behaviour.

Simultaneously, we recognise that frontal inflow does not appear in all previous studies (e.g. Filippi et al., 2018). Furthermore, in our simulation of the SCQ fire, the frontal inflow also shows spatial variability, occurring predominantly in the northern section of the front (Fig. 8). Hence, to further understand the influence of pyro-convection on fire behaviour, we need to know
under which conditions (extreme) wildfire events can create frontal inflow and subsequently how the changes in airflow at the rear and in front of the fire affect the fire behaviour.

## 4.2 Fire-modified boundary layer

We studied the impact of the fire-modified winds (Fig. 13) on the surrounding thermodynamic structure of the atmospheric boundary layer to investigate whether the SCQ fire counteracted the cooling, moistening and stabilisation of the atmosphere at
night ahead of itself. When possible, these changes could explain the continued burning throughout the night of the SCQ fire. We found no warming or drying of the atmosphere at the surface due to the descending motions (Fig. 10). However, we did find two opposite changes in the boundary layer structure ahead of the fire (Fig.11). Within the first 2 km ahead of the fire, we found a 40 to 60% deepening of the neutral boundary layer. Further ahead, between 2 and 4 km ahead of the fire, we observed a 50% decrease in boundary layer height.
Although the boundary layer height is not directly linked to fire behaviour, it is connected to plume growth. Generally, a higher well-mixed boundary layer is beneficial to plume growth. This suggests that when a fire, such as the SCQ fire, increases





the boundary layer height ahead, it makes it easier for a fire to maintain a convective plume while advancing. We know from observations (e.g. Castellnou et al., 2022), that pyro-convection generally increases fire spread rates. Hence, we hypothesise that by modifying the boundary layer ahead, a fire can more easily sustain a convective plume, which would then promote
continued burning throughout the night, despite worsening burning conditions (i.e., cooling and moistening). The opposite applies when the boundary layer height is decreased; in that case, it becomes harder to sustain a convective plume, which is expected to decrease the ability of a fire to continue burning throughout the night.

This hypothesised impact on the fire behaviour only applies to fires that spread in the same direction as their plume, since that is the region where the boundary layer is modified. Hence, the hypothesis applies to the SCQ fire, since both the fire and
plume spread eastward, which corresponds with the area where the boundary layer structure was modified in the simulation (Fig. 12). Visual observations suggest that the hypothesis is a plausible explanation for the continued burning until midnight by the SCQ fire, as the fire not only burned until midnight (22 UTC) but also maintained a convective plume throughout the night (Fig. 1a). With this explanation, we assume that the SCQ fire continually deepens the boundary layer ahead as it advances forward.

A further complicating factor for predicting the impact of the modified boundary layer structure on the fire behaviour is the formation of a surface inversion throughout the night. The surface inversion is not yet present in this case study as we focus on the period (19–20 UTC) surrounding the shift from a positive to a negative surface heat flux (19:41 UTC). Consequently, there has been insufficient time for the formation of a surface inversion in this case study. We expect that the formation of a surface inversion in the environment of the SCQ fire would limit the ability of the fire to maintain a convective plume and
subsequently limit the extent to which the pyro-convection can alter the thermodynamic structure of the atmosphere ahead of the fire. Simultaneously, it is unknown how the fire-induced circulation would affect the surface inversion ahead of the fire. Hence, to test the hypothesis, further studies using a two-way fire atmosphere coupling are needed to investigate the extent to which a modified boundary layer structure explains continued nighttime burning, for example, during the SCQ fire.

### 4.3 Strengths and limitations of in-plume sounding measurements

We specifically chose the SCQ fire as a case study for its availability of both fire-behaviour observations and pyro-convection measurements (CFRS, 2022; Castellnou et al., 2022). For this fire, in-plume soundings were released that measured the thermodynamic structure ($\theta_v$, $q$) inside the plume and the environment during their ascent and descent, respectively. The ability to validate the environment in which the plume was simulated was essential to our validation. The validation revealed an overestimation of the simulated inversion height. Using the environmental measurements of the sounding, we could identify the
underlying cause of the overestimation.

Simultaneously, we recognise the limitations of sounding measurements. It is impossible to control the exact measurement locations as the path of the soundings is controlled by the ambient and in-plume turbulent structures. Furthermore, the spatial and temporal extent of the sounding measurements is limited to a single timeseries and a single measurement per altitude. Lastly, the soundings are intended for atmospheric measurements, which means that they do not provide quantitative measure-
ments of the fire behaviour.





Other tools, such as Doppler radars (e.g. Banta et al., 1992; Clements et al., 2018; Roberts et al., 2024) and Unmanned Aerial Vehicles (UAVs; Kiefer et al., 2012; Bailon-Ruiz et al., 2022), would therefore complement the sounding measurements. UAVs can be used for in-plume measurements of temperature and moisture, similar to soundings, but when equipped with infrared sensors, they can also measure fire behaviour. Doppler radars measure the kinematic plume structure, which complements the
sounding measurements of $\theta$ and $q$. Besides the complementary measurements, depending on their usage, both the Doppler radar and the UAVs provide a larger spatial and/or temporal coverage. Hence, under ideal circumstances, these tools are combined to get a complete set of measurements capturing wildfire-atmosphere interactions: the thermodynamic plume structure (soundings), the kinematic plume structure (Doppler radars) and the fire behaviour (UAVs).

However, Doppler radars and UAVs are relatively expensive and complex to operate compared to soundings. Consequently,
Doppler radars and UAVs are primarily used in dedicated campaigns, whereas the accessibility of the soundings allows for operational usage by firefighters (Ribau et al.). The operational usage means that soundings have the potential to capture a broader range of wildfire-induced plumes, inherently also capturing a wider variety of meteorological conditions. Therefore, we find that soundings offer a baseline opportunity for validating (large-eddy) simulations of pyro-convection. When feasible, we recommend complementing these measurements with additional tools that capture the kinematic plume structure and fire
behaviour, such as Doppler radars and UAVs, to extend the validation possibilities.

### 4.4   Implications for future research

To address the open questions raised in the discussion above, a different methodology is required compared to this case study. In this case study, we chose to implement the SCQ fire as a stationairy area of elevated heat and moisture fluxes. This simplification of the fire behaviour enabled us to use temporal averages to determine whether the fire consistently modified the
surrounding wind (e.g. Fig. 9), $\theta$ and $q$ fields (e.g. Fig. 10). Although this simplification allowed for a focus on the impact of the SCQ fire on the atmosphere, the goal of this case study, it inhibits further study into the effects of the altered atmosphere on the fire spread.

Simultaneously, we used ERA5 boundary conditions to recreate the meteorological conditions of the SCQ fire as realistically as possible to ensure a realistic impact of the simulated fire on the atmosphere. The downside of atmospheric realism is that
the meteorological processes, such as advection and surface fluxes, change simultaneously and continuously over time, making it impossible to isolate the processes governing the wildfire-atmosphere interactions. Consequently, identifying the conditions needed for a fire to create a fire-induced circulation ahead of itself is impossible with the simulation setup of this case study.

Hence, future work could take two opposite directions: either increasing model complexity by including a two-way coupled fire spread model or decreasing complexity by using idealised meteorological conditions. The first pathway enables further
understanding of how the modified atmosphere affects the fire spread behaviour. In contrast, the second is focused on controlling all parameters of the simulation to isolate the processes governing wildfire-atmosphere interactions. Our future work will follow the second path, where we will focus on systematically quantifying and subsequently parameterising the change in rear and frontal inflow. With this parameterisation, we aim to include the effects of pyro-convection on the surface winds in operational fire spread models.



## 5 Conclusions

The objective of this study was threefold: (1) replicate the measured in-situ thermodynamic structure of wildfire plumes using LES; (2) study the proposed impact of pyro-convection on the surrounding wind patterns by previous studies; and (3) analyse the impact of the fire-driven wind patterns on the thermodynamic structure of the atmospheric boundary layer.

The comparison with visual observations and in-plume sounding measurements of the SCQ fire showed that MicroHH is a suitable tool for studying the impacts of pyro-convection on the surrounding kinematic and thermodynamic structure of the atmosphere. The simulated shape and in-plume $\theta_v$ matched the observations. The main discrepancy was the overestimation of the ambient atmospheric boundary layer height by 0.4 km, but this was caused by the ERA5 boundary conditions and does not reflect the performance of MicroHH.

The in-depth analysis of the wind patterns revealed that pyro-convection accelerates the rear inflow, matching the hypothesised primary impact of pyro-convection on the surrounding airflow. However, we also found the formation of a fire-induced circulation ahead of the SCQ fire, which consists of convection within the plume, downdrafts ahead of the fire, and frontal inflow into the fire. We argue that there are two pathways through which the fire-induced circulation can affect the fire behaviour. Firstly, the frontal inflow could counter the accelerated rear inflow, and its presence is associated with the occurrence of fire-generated vortices and long-range spotting. Secondly, we found that the fire-induced circulation modifies the thermodynamic structure of the boundary layer ahead of the fire. The circulation causes simultaneous deepening and thinning of the boundary layer in the 4 km ahead, thereby disrupting the transition from the convective daytime to a stably stratified nighttime boundary layer. This disruption provides a plausible explanation for the sustained nighttime burning during the SCQ fire. Therefore, we argue that the primary impact of pyro-convection on wildfire behaviour depends on the combined effect of changes in wind patterns at the rear and in front of the fire (revised hypothesis), rather than solely on the acceleration of the rear inflow (original hypothesis).

*Code and data availability.* The sounding data and the synoptic weather station data for the SCQ fire are published by Ribau et al. (2022). The simulation settings, along with the surface boundary conditions, are available at https://doi.org/10.5281/zenodo.6433389. This repository also includes the fire perimeter shown in Fig. 1.

## Appendix A: Validation of the periodic boundary conditions

We used periodic boundary conditions for the simulations with MicroHH, allowing air recirculation throughout the domain. It is a commonly used technique to simulate convective boundary layers with turbulence-resolving large-eddy simulation tools such as MicroHH. However, when adding a fire as a source of heat and moisture, we risk that the moisture and heat released by the fire circulate through the domain, subsequently modifying the thermodynamic environment in which the plume develops. To prevent this unrealistic impact on the thermodynamic environment of the plume, we extended the domain eastward. Figures A1 and A2 confirm that the extended domain prevents this unrealistic behaviour. At all four borders, the vertical profiles of the




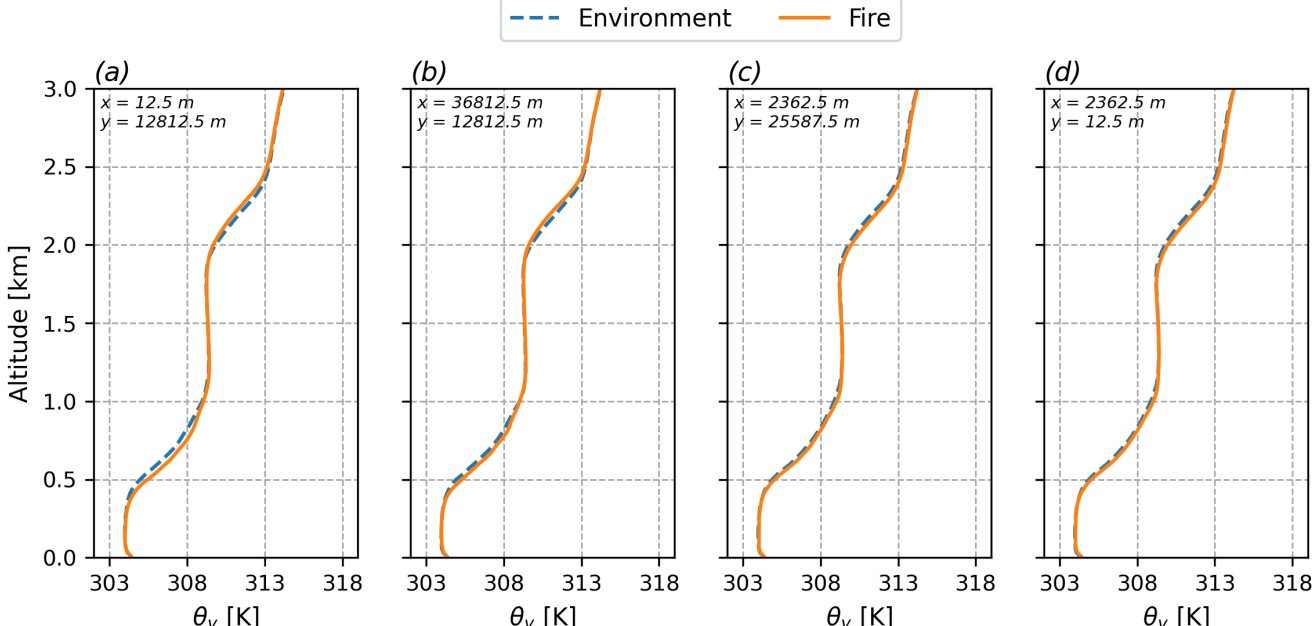

**Figure A1.** The averaged vertical profiles of $\theta_v$ between 19 and 20 UTC for the simulation without (ref-run) and with fire (fire-run) at the four borders of the domain: western **(a)**, eastern **(b)**, northern **(c)**, and southern **(d)**. The x and y values indicate the exact location of the vertical profiles following the coordinate system shown in Fig. 3.

virtual potential temperature and specific humidity do not significantly differ between the simulations with (i.e. fire-run) and without (i.e. ref-run) a fire. Hence, the periodic boundary conditions do not significantly impact the results of this study.

*Author contributions.* TR performed the LES simulations and subsequent analysis and wrote the paper. CH set up the initial setup of the simulations and provided regular feedback on the ongoing analysis and paper writing. All authors assisted with the conceptualisation of the research and acted as internal reviewers.


*Competing interests.* The authors declare that they have no conflict of interest.

*Acknowledgements.* This project received funding from the Union Civil Protection Mechanism of the European Union under grant agreement No 101140363 (EWED). Hence, this project is funded by the European Union. Views and opinions expressed are however those of the authors only and do not necessarily reflect those of the European Union or the European Commission-EU. Neither the European Union





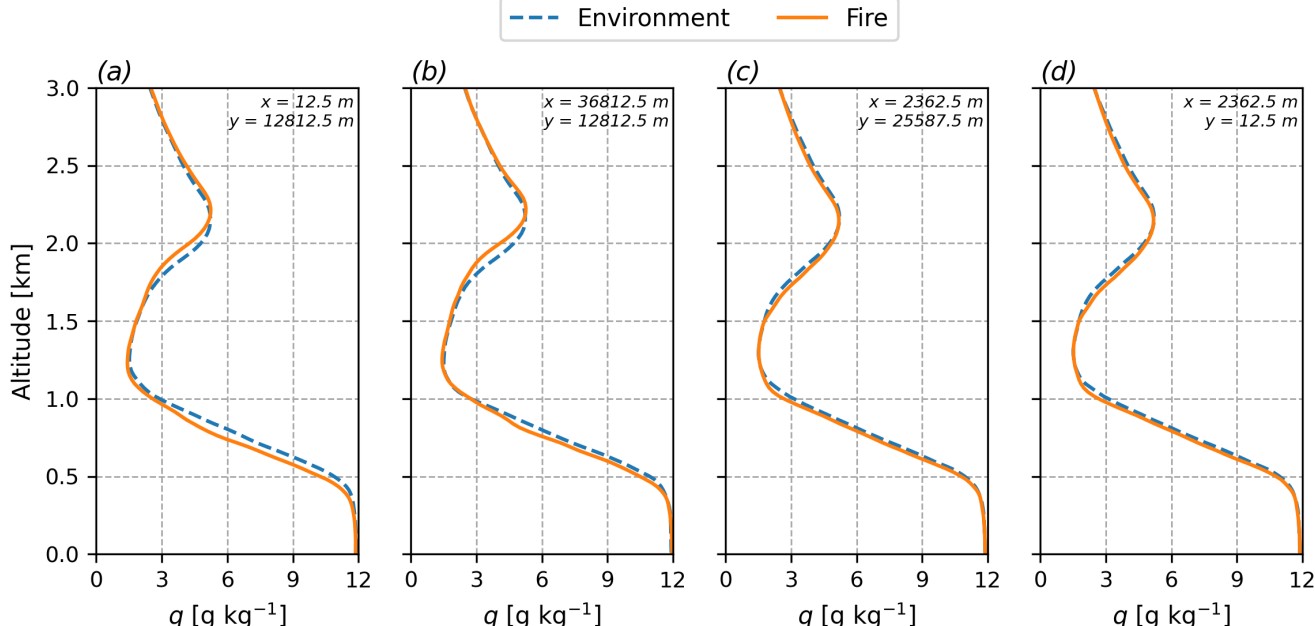

**Figure A2.** The same as in Fig. A1, but for $q$ instead of $\theta_v$.

nor the granting authority can be held responsible for them. Furthermore, we acknowledge the use of AI for coding suggestions (GitHub Co-Pilot) and grammar checking (Grammarly).



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
