# Peer review of "Wildfire-atmosphere interactions during the Santa Coloma de Queralt fire: the development of a fire-induced circulation"

_EGUsphere, 2025_

## Referee Comment (RC2)

**Roelofs et al 2025 manuscript evaluation**

**1 – General comments**

The manuscript presents a new method of evaluating LES simulations of wildfires by using sounding instruments (it is not explicitly said, but it seems to be a radiosonde carried by a weather balloon).

It is focused on studying the atmospheric circulations caused by pyro-convection and how these circulations impact the thermodynamic structure of the plume.

**2 – Errors and typos**

Line 71: it should be "(Ribau et al., 2025)" instead of "(Ribau et al.)".

Line 92: it should be "a sounding was" instead of "a sounding as".

Line 95: the descending profile is said to be "not shown" but Fig. 5a claims: "The vertical profiles, measured during the descent ...".

Line 112: ERA5 is retrieved at a sub-region not centrered on a single point (you actually show this in Fig A1). Provide the bounding box.

Line 136: should it be  $kW/m^2/s$ ?

Line 169: in Line 95 it is said "between 0.2 and 1.4 km" while in L169 it's 0.3 and 1.5 km.

Line 411: same issue with the reference of "(Ribau et al.)".

**Figure 3:**

There seems to be an error in Figure 3 that states that the variable shown is the normalized heat flux while showing units of K.m.s-1. Either show the actual heat flux (most likely in kW.m2.s-1) or show the normalized flux without units. Similarly, in the caption of Fig. 3, the heat flux is in kW.m-2, I believe it should be in kW.m-2.s-1.

Also, provide the x and y axis of the zoomed-in box.

**3 – Detailed comments**

It is unclear what trying to be validated. The proposed approach to validate a LES simulation, where the fire is static, with a single sounding instrument seems ambitious. It would be better to talk about "evaluation" here.

The LES simulation is performed on an idealized set of conditions (no topography, no fuel, no microphysics schemes). This should be clearly stated. Furthermore, there is no coupling between the fire propagation (static heat flux) and the atmosphere.

The simulation parameters could be summarized into a single table.

A list of detailed comments can be found below:

Line 16: To be rephrased. A fireline intensity greater than 10 kW/m is just one of the indicators leading to EWE, other indicators include large values of RoS, Spotting, or plume dominated fire, see Table 2 from Tedim et al. 2018.

Line 84: provide the approximate ignition point if available (or add it to Fig 1).

Line 86: reference the work from Castellnou or Ribau when defining oPyroCu.

Line 89: reference the work supporting that expected behaviour.

Line 116: Is the sea breeze captured by the weather station? If so, provide the data and compare it to the ERA5 reanalysis.

Line 126: provide a reference supporting the claim "this is a common challenge..."

Line 133: Provide the stretch factor and the last cell height.

Line 136: Provide the equation for the moon shape. When is the fire started? How long is the heat flux provided?

**Figure 3**

- It would be clearer if the scale is respected (increasing the x-axis or decreasing the y-axis) to better represent the aspect ratio of the numerical domain.

**Figure 4:**

- it appears in Fig4d that the radiosonde is launched behind the fire injection area (at x=1.9km while the fire seems to start at 2.1km) while on Fig 1a it appears that the radiosonde is launched at a flank of the fire. What is the explanation?
- Please provide a scale for Figs 4 a,b,c.
- Please provide the Y-Z plan view of the plume and the lateral displacement of the radiosonde in that plane.
- Why is there no over shooting in your simulation?

Line 186: Where is the 30% coming from? What are you comparing?

**Figure 5:**

- Instead of "environment" use a different word like "ambient conditions" (used in L189 by the way), "undisturbed air", "background conditions" or other one.
- Include the measurement error (standard deviation or other meaningful metric) from the sounding as horizontal bars or shaded error band.

Line 221: There are also downdrafts in the west section, see Fig 6b.

Line 263: The maximum updraft of 32 m/s seems more realistic. Could you add a figure showing the distribution of updrafts during the 19-20 interval? You can also reference in the discussion some plume observations from Lareau et al, Clements et al and their values to compare with your simulation values.

**Figure 7:**

- How do you calculate the mass flows? Which density are you using?

Lines 243-252: The whole discussion on spot fires seems to be out of scope of this manuscript and the aims of the article.

The spread of spot fires and their coalescence likely depend on their distance to the main fire front:

- if a spot fire appears on the northern part of the front and outside the red square, it will likely spread in a north-east direction (considering the fire is static and there is no interaction between the main fire front and the spot fire).
- If a spot fire appears on the northern part of the front and inside the red square it will likely spread towards the main fire front (negative meridional wind velocity and almost null zonal wind velocity).

Further information is required on the reported spot fires during the SCQ event to make this paragraph useful.

**Figure 9:**

- The maximum average vertical wind velocity of 6 m/s seems very low considering the relatively high fluxes.

Line 300: Can you provide a reference for this method of calculating the ABL height?

Section 4.1: The discussion could be enhanced by including references to the numerical study from Cunningham (2005) ("Coherent vortical structures in numerical simulations of buoyant plumes from wildland fires"), the Lidar analysis from Lareau and Clements (2017) ("The mean and turbulent properties of a wildfire convective plume") where the inflow is clearly visible in their Fig. 5. The fire-induced circulation ahead of the fire front seems to have been observed experimentally. It seems incoherent to talk here about fire behaviour when the fire is modelled as a static heat flux.

Line 450: this claim seems to be in contradiction with the last paragraph of section 4.2 (lines 380-388). In particular, line 382: "Consequently, there has been insufficient time for the formation of a surface inversion in this case study".

---

## Author Comment (AC1)

*We thank Jean-Batiste Filippi for his time and effort in providing this thorough review of our paper. Below in blue & italic, we elaborate on how we provide a detailed response to all feedback, including our approach to include the feedback to improve the paper.*

This manuscript presents a Large-Eddy Simulation (LES) study of the 2021 Santa Coloma de Queralt (SCQ) fire using MicroHH, with the novel inclusion of in-plume radiosonde data for validation. The objective is to examine how pyro-convection modifies near-fire wind patterns and boundary-layer structure, but also the evaluation of Micro-HH and using radio soundings.

The topic is scientifically significant, addressing the mechanisms behind sustained nighttime burning and extreme fire behaviour. The work demonstrates good numerical design and physical interpretation; in particular, the use of radiosonde profiles to validate the plume structure is original and valuable for the field, asn well as comparing it to NWP.

Scientific relevance can therfore be high. The study advances quantitative understanding of fire-induced circulations and provides evidence that frontal inflow, rather than rear-inflow enhancement alone, may governs plume–atmosphere coupling. The use of an LES code not originally developed for wildfire problems shows capability and will interest both fire and boundary-layer communities.

**Major remarks:**

**Scope of "validation". The paper repeatedly refers to validation, but it is not fully clear what is validated—MicroHH as a model, the fire setup, or the specific thermodynamic representation. Maybe just a "Comparison"or investigation. A clearer statement that this is a comparative test against a single radiosonde, not a formal model validation, would help.**

*The goal of comparing the simulation output against the radiosonde measurements was to focus on validating the thermodynamic structure inside the simulated pyro-convective plume. It was not intended to function as a full model validation of MicroHH.*

*To improve the clarity of the paper regarding this topic, we will change the terminology to evaluation/comparison, as these terms cover our goal with the radiosonde measurements: to compare how well MicroHH can replicate the radiosounding measurements.*

Furthermore, with the following 3 statements in the introduction, methods and methodology, we expect that we have clarified the goal of the comparison within the paper:

1. **Introduction: (L74):** *the first objective of this study is to demonstrate the ability of LES to reproduce in-plume soundings of thermodynamic plume structures during extreme wildfire events.*

2. **Methods (L107-109):** *Consequently, we will use the radiosonde measurements to evaluate both the simulated pyro-convection and the ambient conditions in which the pyro-convection was simulated. For the evaluation, we will focus on comparing the observed and simulated temperature profiles. Furthermore, we will also compare the measured wind speed in the convective plume with the simulated wind speed.*

3. **Results (L212):** *To evaluate the simulated pyro-convection between 19 and 20 UTC, we compared the simulated plume shape and vertical profile of the virtual potential temperature ($\theta_v$) and wind speed (U) with visual observations (Fig. 5) and the in-plume radiosonde (Fig. 6).*

Please note that we added a new evaluation metric: wind speed. An elaborate discussion on the choices in evaluation metrics is provided as a response to the third comment below.

**Fire representation. The fire seems to be implemented as a dynamic heat-flux patch maybe with explicit combustion or spread, but it seems very vague or unclear in the current redaction, do you have any isochrones, fuel maps, orography, it appears not, as well as boundary condition, it is perfectly OK, but if it is a somehow idealized fire it should be clearly presented as such. Also this simplification should be justified earlier and clearly separated from coupled fire-atmosphere modelling claims. And if you have, it would be useful to provide basic the actual parameters of the assumed fuel type and flux intensity and to state whether topography was flat or taken from ERA5.**

To improve the clarity on the fire representation in our study we will split section 2.2. into two subsections, 2.2.1 & 2.2.2. Section 2.2.1 specifically addresses the fire implementation in MicroHH, while section 2.2.2 serves as a description of the meteorological boundary conditions.

Regarding the fire implementation in section 2.2.1, to clarify the choices we made for our case study of the SCQ fire, we will explicitly state the following:

- We implemented the SCQ fire as a stationary, moon-shaped, constant heat flux patch. Hence, we do not simulate combustion or the spread of the fire.

- o As you suggested, we will directly introduce the justification for the stationarity of the implemented fire, which is that: We assume that the movement of the fire is negligible from an atmospheric perspective as the fire rate of spread is significantly lower than the wind speed.
- The dimensions and the moon-shaped form of the heat flux patch are based on observations of the Catalan Fire and Rescue service, including the observed fire perimeters, which are shown in Figure 1a as hourly fire perimeters.
- The heat flux is based on the dominant fuel type (pinus halepensis), ROS, and fuel energy content based on Byram's equation for the FLI (Byram, 1959).
  - o This includes a 50% correction for the radiative heat loss and a correction for the fine fuel moisture content (similar, albeit slightly different to Lareau & Clements, 2016)

**Boundary conditions. ERA5 forcing and the periodic lateral boundaries may influence inversion height and plume recirculation. A short sensitivity test or discussion (possibly moved from the Appendix) should quantify the expected impact.**
We will move figure A1 up to the methods, creating figure 4, which shows the profiles of the virtual potential temperature at all four lateral boundaries of the domain for both the *ref-run* and *fire-run*. Comparing the two simulations shows that there is no impact of the lateral recirculation on the boundary layer structure (including inversion height). Hence, we conclude that the chosen domain size is sufficiently large to prevent any unwanted recirculation of the fire-induced plume.

The analysis of the specific humidity stays in Appendix A (Fig A1). While it does not introduce new insights beyond Figure 4, it confirms the findings regarding the insignificant impact of lateral recirculation due to the periodic boundary conditions.

**Physical metrics. Beyond potential temperature, additional diagnostics (e.g., CAPE, wind profile, potential temperature) could strengthen the interpretation of the radiosonde comparison and the discussion of plume dynamics.**

We agree with your suggestion, those measurements could strengthen the interpretation of the radiosonde comparison. Below we elaborate per parameter why we will or will not include them in the

- **Cape:** is possible when we have a full environmental profile, but that one misses essential data on the near-surface properties of the atmosphere (Fig. 6a).

- **Wind speed profile:** We think this is a great suggestion, and we will add a comparison between the simulated in-plume wind speed and the observed in-plume wind speed. We do not compare the ambient wind speed profile because the measurements during the descent were taken at a negative vertical velocity of 8 to 9 m/s, rendering them unreliable for the comparison.
- **Specific humidity:** Although reliable, it is not of big interest to our case study. During our period of interest, 19 to 20 UTC, a dry convective plume was observed (i.e. no pyro-clouds), and MicroHH also produced a dry convective plume. Hence, the simulation is sufficiently dry. Subsequently, comparing the simulated moisture amount with the observations will not add value to our case study.

To clarify these decisions on what parameters we include, we will add the following statement to section 2.1 after introducing the radiosonde measurements and the extent of the comparison (see comment above): *For the evaluation, we will focus on comparing the observed and simulated temperature profiles. Furthermore, we will also compare the measured wind speed in the convective plume with the simulated wind speed. The measured wind speed during the descent of the radiosonde in the surroundings of the plume and the measured specific humidity are not used for the comparison. We consider the wind speed measurements during descent unreliable due to the high descent speed of the radiosonde (8 – 9 m s$^{-1}$). The specific humidity measurements, on the other hand, are reliable but redundant for evaluating our simulation, as both observations and the simulation indicate a dry convective plume (i.e., no pyro-clouds). Hence, the simulation was sufficiently dry.*

**Figures. Some figures (e.g., Fig. 3–5) would benefit from clearer units and labels— particularly for "normalised flux", velocities (m s$^{-1}$), and altitude scales.**
Thanks for pointing out the unclarity in the units. Below we discuss our improvements per figure(s):

- **Fig 3. (Fig 2 in the revised manuscript):** We recognise the mistake with the units; a normalised unit should not have a unit. However, considering also the feedback from Anonymous reviewer 1, We will replace the normalised heat flux with the actual heat flux in kW/m2 to match the units used throughout the text. Additionally, we will rescale the ratio of the axis to better represent the actual distance. (The x-axis represents a longer spatial distance in km than the y-axis, but currently is equally long as the y-axis in the figure, which hinders effective interpretation.)
- **Fig 2,4, and 5 (3-6 in the new manuscript):** To improve the clarity on the 'Altitude', we will replace the 'Altitude' label on the y-axis with 'Altitude AGL' to refer to the altitude above ground level (AGL). Additionally, we will add a clarification in the

methodology (section 2.2.1) about how elevation works in our simulation. We do not account for local topography, treating the terrain as a flat plain. However, we do consider the overall elevation of the region where the SCQ fire occurred using the ERA5 pressure field. Hence, the altitude in the simulation is equal to the altitude above ground level.

**Minor issues**

**– Define clearly "frontal inflow" and "rear inflow" on first use**.

We modified L45 and L48 to include an explicit definition of the rear and frontal inflow. Please note that we added Lareau & Clements (2017) as an additional source at the end of L48, per a suggestion from another reviewer.

*Old:*

*L45 : Despite the variability, the LES studies agree with the theory proposed by Potter (2012b) that wildfires can accelerate the upwind inflow (Coen et al., 2013; Peace et al., 2016; Filippi et al., 2018).*

*L46-?...: In addition to the accelerated upwind inflow, the simulations by Coen et al. (2013) and Peace et al. (2016) show that wildfires can also modify the downwind airflow, creating frontal inflow, a feature of pyro-convection regularly used operationally to set backing fires.*

*New:*

*L46: Despite the variability, the LES studies (Coen et al., 2013; Peace et al., 2016; Filippi et al., 2018) agree with the theory proposed by Potter (2012b) that wildfires can accelerate the rear inflow, defined as the entrainment of air into the plume from behind the flaming zone.*

*L48-52?: In addition to the accelerated upwind inflow, the simulations by Coen et al. (2013) and Peace et al. (2016) show that wildfires can also modify the downwind airflow. This creates frontal inflow, where air is entrained into the fire from ahead of the flaming zone. The creation of frontal inflow by pyro-convection is regularly used operationally to set backing fires and matches with Doppler measurements of convective wildfire plumes, which reveal significant frontal inflow into the fire (Banta et al., 1992; Lareau and Clements, 2017; Roberts et al., 2024)*

**– Clarify whether "ERA5" or "ERA-5" is used consistently.**

*Thanks for noticing the inconsistency. We chose ERA5 and checked the consistency throughout the paper.*

**– Proofread for minor grammatical errors and duplicated references.**

Overall the manuscript is scientifically sound and offers a significant contribution to the understanding of wildfire-atmosphere coupling. It would merit full review and likely publication after major revisions aimed at clarifying the methodological scope (validation vs. comparison), documenting the fire setup, and tightening figure presentation. Given these strengths and the importance of the dataset, I recommend accepting it for external review

**References:**

Lareau, N. P. and Clements, C. B.: Environmental Controls on Pyrocumulus and Pyrocumulonimbus Initiation and Development, Atmospheric Chemistry and Physics, 16, 4005–4022, https://doi.org/10.5194/ACP-16-4005-2016, 2016

Byram, G. M.: Combustion of Forest Fuels, in: Forest Fire: Control and Use, edited by K. P. Davis, pp. 61–89, McGraw-Hill, New York, 1959

---

## Author Comment (AC2)

*We thank Anonymous reviewer 1 for their time and effort in providing this thorough review of our paper. Below in blue & italic, we elaborate on all feedback, including our approach to include the feedback to improve the paper.*

**1 – General comments**

The manuscript presents a new method of evaluating LES simulations of wildfires by using sounding instruments (it is not explicitly said, but it seems to be a radiosonde carried by a weather balloon).

*You are right here, it is indeed a radiosonde carried by a weather balloon, albeit a smaller helium filled balloon then typically used by operational weather services. The initial methodology was provided by Castellnou et al. (2022). Hence, to improve the clarity on this topic, we will add an explicit reference to Castellnou et al. (2022), as they initially presented the in-situ data and measurement technique. To do so, we will add the following line in section 2.1: A description of the equipment and measurement techniques is provided by Castellnou et al. (2022).*

It is focused on studying the atmospheric circulations caused by pyro-convection and how these circulations impact the thermodynamic structure of the plume.

**2 – Errors and typos**

Line 71: it should be "(Ribau et al., 2025)" instead of "(Ribau et al.)".

*Thanks for the observation. We will fix the reference.*

Line 92: it should be "a sounding was" instead of "a sounding as".

*Thanks for noticing, we will correct the grammar following your suggestion.*

Line 95: the descending profile is said to be "not shown" but Fig. 5a claims: "The vertical profiles, measured during the descent ...".

*We recognise the apparent paradox between L95 and Fig. 5a.*

*The intent in L95 was to clarify that the descending profile of the rising speed during the descent was not shown in Figure 1, in contrast to the rising speed during the ascent of the sounding, which is shown in Figure 1b. Nevertheless, we still want to use the data for comparing the simulated environment with the observed environment, as is done in Fig. 5a.*

*In the updated manuscript, the methodology was restructured in response to your and the other reviewers' feedback. Consequently, there is no need anymore to indicate that the descending velocity is not shown in Fig. 1. Hence, the apparent paradox is not present anymore in the text.*

Line 112: ERA5 is retrieved at a sub-region not centrered on a single point (you actually show this in Fig A1). Provide the bounding box.

We understand the apparent paradox regarding the ERA5 boundary conditions. This is related to how MicroHH is used.

The periodic boundary conditions of MicroHH, the LES tool used in this study, mean that the air exiting at the west boundary re-enters the domain at the east boundary. To make the recirculation physically make sense, the domain, even though it is a 3D space, represents a single point on the ERA5 grid. This means that the full domain is forced by a single ERA5 grid point.

We recognize that in the initial methods, it was unclear for the reader that MicroHH uses periodic boundary conditions when introducing the ERA5 boundary conditions, hence we adjusted line 113-114 from:

*The ERA5 data is retrieved at the centre of the SCQ fire (latitude: 41.51775°, longitude: 1.494428°)*

Into (line 179 in the restructured methodology section):

*We retrieved the ERA5 boundary conditions at a single point, the centre of the SCQ fire (latitude: 41.51775°, longitude: 1.494428°), since MicroHH uses periodic boundaries.*

Line 136: should it be kW/m2/s ?

The unit was a formatting accident; we did not mean the normalised heat flux to have units. However, considering your feedback and the feedback of Jean Baptiste Filipe, we decided to avoid complexity and just show the actual fire intensity in kW/m2 in Figure 3 **(Note, this is figure 2 in the revised manuscript)**. This unit is consistent with the text.

- **Note:** In K m-1 s-1, the seconds are explicitly mentioned to indicate a heating rate, but in kW m-2, the s-1 is included in the Wattage (W), which represents J/s, hence we use kW m-2, not kW m-2 s-1

Line 169: in Line 95 it is said "between 0.2 and 1.4 km" while in L169 it's 0.3 and 1.5 km.

Thanks for spotting the mistake; the correct values are 0.3 and 1.5 km, as used in Line 169, which matches with Figure 1b. We accidentally forgot to update the values on Line 95 during the manuscript revision before submission.

Line 411: same issue with the reference of "(Ribau et al.)".

Thanks for the observation. We will fix the reference.

Figure 3: There seems to be an error in Figure 3 that states that the variable shown is the normalized heat flux while showing units of K.m.s-1. Either show the actual heat flux (most likely in kW.m2.s-1) or show the normalized flux without units. Similarly, in the caption of Fig. 3, the heat flux is in kW.m-2, I believe it should be in kW.m-2.s-1. Also, provide the x and y axis of the zoomed-in box.

Regarding the units and the type of heat flux shown, please see our response above to your comment on line 136.

To improve the figure, we will add the x- and y-labels to the insert.

**3 – Detailed comments**

It is unclear what trying to be validated. The proposed approach to validate a LES simulation, where the fire is static, with a single sounding instrument seems ambitious. It would be better to talk about "evaluation" here.

This remark matches with the feedback from the other reviewer, Jean-Baptiste Filippi, we recognise that 'validation' results in too high expectations compared to what we can actually do. Hence, we will use the terms 'comparison' and 'evaluation' instead of Validation.

The goal of the comparison is to assess how well MicroHH can represent the thermodynamic structure within the pyro-convective plume relative to in-plume measurements. In the revised manuscript, we have the following three statements in the introduction, methodology (2.1) and results (3.1) to clarify what the goal of the comparison is:

1. **Introduction:** (L74): *the first objective of this study is to demonstrate the ability of LES to reproduce in-plume soundings of thermodynamic plume structures during extreme wildfire events.*
2. *Methods* (L107-109): *Consequently, we will use the radiosonde measurements to evaluate both the simulated pyro-convection and the ambient conditions in which the pyro-convection was simulated. For the evaluation, we will focus on comparing the observed and simulated temperature profiles. Furthermore, we will also compare the measured wind speed in the convective plume with the simulated wind speed.*
3. **Results** (L212): *To evaluate the simulated pyro-convection between 19 and 20 UTC, we compared the simulated plume shape and vertical profile of the virtual potential temperature ( $\theta_v$) and wind speed (U) with visual observations (Fig. 5) and the in-plume radiosonde (Fig. 6).*

The LES simulation is performed on an idealized set of conditions (no topography, no fuel, no microphysics schemes). This should be clearly stated. Furthermore, there is no coupling between the fire propagation (static heat flux) and the atmosphere.

Your understanding is fully correct. This also aligns with the other reviewer's feedback, which found the current description too implicit. To solve this issue, we will separate 2.2. into 2 sections, one being the fire implementation (2.2.1) and the other the meteorological boundary conditions (2.2.2).  Additionally, we will expand the methodology in section 2.2.1 to explicitly state the simplification separate from any claims on wildfire-atmosphere interactions. Furthermore, the fuel type and properties used to derive the fire intensity used in our simulation will be added to this section.

The simulation parameters could be summarized into a single table.

We will add a table to section 2.2 that includes the general simulation parameters valid for both the ref- and fire-run (Table 1).

**A list of detailed comments can be found below:**

Line 16: To be rephrased. A fireline intensity greater than 10 kW/m is just one of the indicators leading to EWE, other indicators include large values of RoS, Spotting, or plume dominated fire, see Table 2 from Tedim et al. 2018.

Thanks for the observation; a nuance is indeed needed here. We changed the first line of the introduction to:

*Extreme wildfire events are defined by fire behaviour that surpasses the extinguishing capacity of fire services, indicated by factors such as high rates of spread, erratic spotting, or fireline intensities exceeding 10 MW m-1 (Tedim et al., 2018). During events with such high fireline intensity, the heat release can trigger significant upward convective motions (i.e. pyro-convection)...*

**Old version:**

*L16: Extreme wildfire events occur when the fire line intensity of wildfires exceeds the extinguishing capacity of the fire service of 10 MW m−1 (Tedim et al., 2018).*

Line 84: provide the approximate ignition point if available (or add it to Fig 1).

We will add the ignition point with the introduction of the Santa Coloma de Queralt fire at the start of section 2.1. We chose not to add it to Figure 1a, as it would overlap with the fire perimeters.

Line 86: reference the work from Castellnou or Ribau when defining oPyroCu.

Thanks for catching the missing source; we will refer here to Castellnou et al. (2022), who proposed the definition of oPyroCu.

Line 89: reference the work supporting that expected behaviour.

We agree that the discussion about the observed plume behaviour in the sentence: *"It is expected that the stabilisation of the atmosphere limited the vertical extent of the plume, inhibiting the plume from reaching the lifting condensation level"* deserves a reference, considering the connection between atmospheric stability and plume height.

Upon re-evaluation of section 2.1, this specific discussion, which connects changes in atmospheric stability to the plume's ability to create pyroclouds, is unnecessary to our paper, as our period of interest, 19 to 20 UTC on the 24$^{th}$ of July 2021 (due to the availability of measurements) covers only a dry convective plume (i.e no pyrocloud formation was observed). This means that the physics regarding the formation/dissipation of pyro-clouds (i.e. moist pyro-convection) is outside the scope of our paper. Hence, we decided to remove the sentence to improve the clarity and conciseness of this section. This ensures the section is focused on the observed persistence of the dry convective plume until midnight despite the worsening burning conditions and arrival of the sea breeze.

Line 116: Is the sea breeze captured by the weather station? If so, provide the data and compare it to the ERA5 reanalysis.

To enhance the discussion on the sea breeze we will add the 10 m wind speed and direction measurements by the weather station as panel **d** in Fig. 1. It shows that combined with the strong increase in RH at 19 UTC of +/- 80 %, the wind also showed a slight backing of 30 to 60 degrees. This is significantly less than the +/- 90 degrees backing visible in the ERA5 boundary conditions.

We will add these observations to the discussion on the quality and impact of the ERA5 boundary conditions in section 2.2.2

- **Note:** A more in-depth analysis of the mesoscale effects of the sea breeze on pyro-convection during the SCQ fire on the 24th of July is currently under review at a different journal (No preprint available, unfortunately).

Line 126: provide a reference supporting the claim "this is a common challenge…"

We will add a reference to the following two sources:

1. Gualtieri, 2021
2. Zuo et al., 2025

Although their focus is on wind energy, they consistently show in two different areas that increased topography reduces the performance of ERA5, indicating that it is challenging for ERA5 to capture complex local topography.

In addition, we connected the discussion on ERA5 performance in complex topography back to the environment in which the SCQ fire occurred.

This will result in the following discussion in section 2.2.2, replacing the claim in Line 126:

*Previous validation studies of ERA5 in complex terrain show that a horizontal resolution of 31 km is often insufficient to resolve small-scale orographic features, leading to significant discrepancies in local winds (Gualtieri, 2021; Zuo et al., 2025). For the SCQ fire, this suggests that the ERA5 grid is too coarse to resolve the coastal mountain range in Catalunya, which acted as a barrier to the sea breeze, delaying its arrival until 19 UTC (CFRS, 2022).*

Line 133: Provide the stretch factor and the last cell height.

We will add this information in the newly created Table 1 (see discussion above)

Line 136: Provide the equation for the moon shape. When is the fire started? How long is the heat flux provided?

In the newly created section 2.2.1 (see discussion above), we will add the following statement to clarify the temporal properties of the implemented fire:

*To implement a fire in MicroHH during the fire-run, we added a stationary moon-shaped area with a constant heat flux of 145 kW m−2 (Fig. 2) between 18 to 20 UTC on top of the ambient fluxes from ERA5 (see Sect. 2.2.2).*

Furthermore, we will add the two quadratic equations that we used to define the moon-shaped fire in our simulation.

**Figure 3**

- It would be clearer if the scale is respected (increasing the x-axis or decreasing the y-axis) to better represent the aspect ratio of the numerical domain.

   *We agree, we will lengthen the x-axis to get an appropriate ratio between the x and y-axis.*

**Figure 4:**

- it appears in Fig4d that the radiosonde is launched behind the fire injection area (at x=1.9km while the fire seems to start at 2.1km) while on Fig 1a it appears that the radiosonde is launched at a flank of the fire. What is the explanation?

Your observation on the sounding launching location in Fig. 1a is correct. With the comparison, we have two big challenges.

1. The location of the radiosonde measurements relative to the plumes locations is unknown as we don't have observations of the exact location of the plume during the sounding.
2. We cannot replicate the exact plume shape of the turbulent plume at the time of the sounding, since turbulence is a chaotic process. Instead we can replicate the average shape during the period of interest, in our case 19 to 20 UTC.

We do know that the sounding moved predominantly eastward and that it measured inside the convective core of the plume between 0.3 and 1.5 km altitude (due to the rising speed exceed 2 m/s).

At this point, we had to make a decision from the modelling perspective: how do we capture the convective core of the plume in our simulation? Our solution was to take a cross-section through the center of the fire in eastward direction, the xz cross-section at y = 12.8125 km shown in Fig 4d as it captures the convective plume in the simulation and matches with the dominant direction of the radiosonde. Hence, in figure 4d, we only show the east-west displacement of the sounding. From this 2D perspective, the radiosonde seems to be launched from behind the fire (Fig 4d), whereas it in reality was launched at the flanks (1a). Two comments below we discuss how we will add also the north-south movement of the radiosonde to the paper.

We recognize that we did not make these decision explicit in section 3.1, so to improve the clarity of the comparison between the sounding and the radiosounding measurements, we will add the following paragraph in section 3.1:

*For the remainder of this comparison, we will compare the radiosonde measurements to the simulation, which has two challenges. Firstly, the location of the radiosonde relative to the plume during the measurements is unknown, as we do not have observations on the exact location of the plume during the sounding. Secondly, we cannot replicate the exact shape of the turbulent plume at the time of the sounding, since turbulence is a chaotic process. Instead, we can replicate the*

*average shape of the plume between 19 and 20 UTC, the period in which the sounding was released. Despite these challenges, we know that the radiosonde moved predominantly eastward (Fig. 1a) and that it measured inside the convective core of the plume between 0.3 and 1.5 km altitude (Fig. 1b). To capture the convective core of the plume in the simulation, we extract a xz cross-section at y = 12.8125 km. This xz cross-section represents an east-west plane through the centre of the fire, matching the predominant eastward movement of the radiosonde. Hence, we will compare the radiosonde measurements with the averaged xz cross-section of the plume between 19 and 20 UTC through the centre of the fire.*

- Please provide a scale for Figs 4 a,b,c.

  The images are derived from video footage captured by cameras from the Catalan Fire and Rescue Service, which are moved and zoomed in and out over time during the SCQ fire. We deem it impossible to accurately retrieve scales for these images. Moreover, this is unnecessary for the goal of these images. The aim is to show the main visible features that were observed qualitatively (e.g. the overshooting) and compare those to the features of the simulated plume rather than using them to assess the size of the simulated plume.

- Please provide the Y-Z plan view of the plume and the lateral displacement of the radiosonde in that plane.

  We infer from your comment that you want to have insight into the full movement of the sounding, meaning both the horizontal and vertical movement. Currently, we showed the vertical movement and horizontal movement in the east-west direction in Figure 4d, but we did not show the horizontal movement in the north-south direction. Adding a Y-Z cross-section would solve this, but such a cross-section does not show a full plume, as the simulated plume develops in the east-west direction.

  This makes it impossible in the Y-Z cross-section to compare the north-south movement of the radiosonde to the average shape of the plume. **Hence, we do not consider a figure of the Y-Z cross-section a beneficial addition to the paper.**

  Instead, **we added the horizontal movement of the sounding in Figure 1a**, with the grey circles indicating horizontal movement outside the convective core of the plume (rising speed < 2 m/s) and the red circles the horizontal movement inside the convective core of the plume (rising speed > 2 m/s).

- Why is there no over shooting in your simulation?

  There is occasional overshooting, but not consistently enough to get included in the average plume outline in Figure 4d. This is discussed in L163: *Occasional overshooting also occurs in the simulation (not shown), but not consistently enough to affect the average plume shape.*

Line 186: Where is the 30% coming from? What are you comparing?

  The estimated 30% mismatch in tilt was an approximation between the horizontal displacement path of the radiosonde between 0.3 and 1.5 km altitude and the estimated tilt on the right border of the convection column boundary (grey outline) shown in Fig. 5d.

  Following the rework and the refinement of section 3.1, we found that this quantitative comparison is no longer needed for the comparison between the sounding and the simulation. A single value to approximate the mismatch in the tilt between observations and the simulation does not add value to the discussion and conclusion of the comparison. Hence, for clarity and conciseness, we removed the quantification.

**Figure 5:**

- Instead of "environment" use a different word like "ambient conditions" (used in L189 by the way), "undisturbed air", "background conditions" or other one.

  We will use ambient conditions.

- Include the measurement error (standard deviation or other meaningful metric) from the sounding as horizontal bars or shaded error band.

  Although we understand the wish for a measurement error, this is impossible for sounding measurements, as they only provide a single measurement per height and per time step.

Line 221: There are also downdrafts in the west section, see Fig 6b.

Thanks for your observation, we agree, we will rectify this in the text.

Line 263: The maximum updraft of 32 m/s seems more realistic. Could you add a figure showing the distribution of updrafts during the 19-20 interval? You can also reference in the discussion some plume observations from Lareau et al, Clements et al and their values to compare with your simulation values.

Thanks you for your suggestion, we will add a probability density function of the instantaneous vertical velocity inside the plume for both the ref-run and fire-run:

[Figure]

As you also comment on Figure 9 (see your comment below), the 32 m/s max updraft speed seems more realistic then the vertical velocities shown in Fig 9c. Nonetheless, both are true for our simulation of the SCQ fire.

In figure 9c, similar to all other figures, we show the averaged vertical velocity between 19 and 20 UTC. Averaging over time means that we smooth out the individual peaks in vertical velocity throughout the hour. Consequently, instead a concentrated peak in vertical velocity, figure 9c shows a smoothed area with relatively high vertical velocity of around 6 m/s.

During the individual timestep in between 19 and 20 UTC (we have output every 60 second for the cross-sections shown in Figure 9c), we of course don't have the smoothing and we get significantly higher values for the vertical velocity, up to 32 m/s. By adding the probability density function (figure above) to the paper, we are able to visualize this difference. Moreover, as you suggested, this helps us to compare the simulated updrafts with observations from Lareau and Clements 2017 and Clements et al. 2018.

**Figure 7:**

- How do you calculate the mass flows? Which density are you using?

We follow the mass conservation definition within MicroHH defined in Eq. 2 in Van Heerwaarden et al. (2017), which uses a reference density that is a function of height only. We will refer to Van Heerwaarden et al. (2017) in the text to clarify how we calculated the mass flows.

Lines 243-252: The whole discussion on spot fires seems to be out of scope of this manuscript and the aims of the article. The spread of spot fires and their coalescence likely depend on their distance to the main fire front:

- if a spot fire appears on the northern part of the front and outside the red square, it will likely spread in a north-east direction (considering the fire is static and there is no interaction between the main fire front and the spot fire).

- If a spot fire appears on the northern part of the front and inside the red square it will likely spread towards the main fire front (negative meridional wind velocity and almost null zonal wind velocity).

Further information is required on the reported spot fires during the SCQ event to make this paragraph useful.

We agree with your assessment. The original intent with the discussion on spot fires in Lines 243-252 was to illustrate the potential relevance of the wind modifications at the eastern border.

However, we recognise that introducing spot fire dynamics without detailed observations of spot fire behaviour during the SCQ event distracts from the primary focus of the results section, which is to quantify the impact of pyro-convection on the surrounding wind patterns. Furthermore, presenting interpretations of fire behaviour (e.g. spotting fires) as "results" is inconsistent with our methodology. We used a stationary fire to isolate atmospheric effects. Hence, our simulation does not provide dynamic fire-behaviour output.

Therefore, we decided to remove Lines 243 to 252 to keep the results section concise and focused.

**Figure 9:**

- The maximum average vertical wind velocity of 6 m/s seems very low considering the relatively high fluxes.

Please, see our response to your comment on line 263 above.

Line 300: Can you provide a reference for this method of calculating the ABL height?

The methodology here is based on the mixed-layer framework, described by Vila-Guerau De Arellano et al. (2015). We will refer Vila-Guerau De Arellano et al. (2015) in text.

**Section 4.1:**

-   The discussion could be enhanced by including references to the numerical study from Cunningham (2005) ("Coherent vortical structures in numerical simulations of buoyant plumes from wildland fires"), the Lidar analysis from Lareau and Clements (2017) ("The mean and turbulent properties of a wildfire convective plume") where the inflow is clearly visible in their Fig. 5. The fire-induced circulation ahead of the fire front seems to have been observed experimentally.
    1.  Lareau and Clements et al. 2017: We agree that this is an amazing observational study that clearly shows the frontal inflow. Hence, we added this reference to section 4.1.
    2.  Cunningham et al. (2005): Their study focused on replicating and understanding the origin of vortices inside a buoyant wildfire plume and the development of wake vortices (also known as fire-generated vortices). In our study, we find a large-scale circulation spanning around 2 km, which does not seem to match any of the circular structures (i.e. vortices) described by Cunningham et al. (2005). We recognise that there may be a relationship between the fire-induced circulation we found and the vortices described by Cunningham et al. (2005), but, based on our work, we cannot draw any conclusions or hypotheses about it, since we did not investigate vortical structures in our simulation. Hence, we left this reference out of the discussion in section 4.1. Nonetheless, we do recognise this study as a very clear example of a highly controlled and simplified simulation setup that we describe in section 4.4. Hence, we will add it there as a reference.
-    It seems incoherent to talk here about fire behaviour when the fire is modelled as a static heat flux.

    We acknowledge the reviewer's concern regarding the distinction between our static modelling approach and the discussion of dynamic fire behaviour.

Upon re-evaluation of section 4.1, we decided to restructure the section to refocus it on the impacts of pyro-convection on the surrounding wind patterns, the objective of our paper. With this restructuring, we removed the connection to the fire behaviour, as this is indeed one step to far compared to the methodology that we used (i.e. a stationary fire).

- o **Note 1:** We also revised the conceptual figure (Fig 15b, the updated figure is inserted below) based on feedback we received when we presented during the Workshop on Numerical wildfire and AI Weather modelling (https://forefireapi.github.io/cargese2025/).  Originally, we drew the fire-induced circulation as a combination of the updraft, downdraft, and frontal inflow, but it was noted that in Figures 9, 10, and 12, the circulation appears separate from the convective plume (1) and downdrafts (3). We agree with that view, so we redraw the fire-induced circulation as a separate entity (4) in the conceptual figure between the convective plume and the downdrafts.

[Figure]

Line 450: this claim seems to be in contradiction with the last paragraph of section 4.2 (lines380-388). In particular, line 382: "Consequently, there has been insufficient time for the formation of a surface inversion in this case study".

Thanks for pointing this out, the current wording indeed causes a contradiction between the discussion and the conclusion. The conclusion should have had more nuance similar to the discussion in section 4.2.

Based on your feedback regarding section 4.1, we also carefully considered what content should be included in the conclusion. We decided to focus the conclusion on the main findings regarding the impact of pyro-convection on the surrounding atmosphere, since this is the main objective of our case study and aligns with the methodology we used. Hence, we removed the hypothetical impact on the fire behaviour of the changed

thermodynamic structure of the atmospheric boundary layer from the conclusion. Consequently, we leave the impact of the changed thermodynamic structure for discussion (section 4.2), where we can examine it from all perspectives, thereby providing the required nuance to our findings.

With this restructuring of the conclusion, we also removed the contradiction from the text.

**References:**

Castellnou, M., Bachfisher, M., Miralles, M., Ruiz, B., Stoof, C. R., and de Arellano, J. V.-G.: Pyroconvection Classificatin Based on Atmospheric Vertical Profiling Correlation with Extreme Fire Spread Observations, Journal of Geophysical Research: Atmospheres, 127, e2022JD036 920, https://doi.org/10.1029/2022JD036920, 2022.

CFRS: Fire Information Sheet: Santa Coloma de Queralt, Tech. rep., Catalan Fire Rescue Service (GRAF unit), 2022.

Gualtieri, G.: Reliability of ERA5 Reanalysis Data for Wind Resource Assessment: A Comparison against Tall Towers, Energies, 14, https://doi.org/10.3390/en14144169, 2021.

Zuo, P., Chen, X., Zhu, L., Zuo, P., Chen, X., and Zhu, L.: Applicability Assessment of ERA5 Surface Wind Speed Data Across Different Landforms in China, Atmosphere, 16, https://doi.org/10.3390/atmos16080956, 2025.

Lareau, N. P. and Clements, C. B.: The Mean and Turbulent Properties of a Wildfire Convective Plume, Journal of Applied Meteorology and Climatology, 56, 2289–2299, https://doi.org/10.1175/JAMC-D-16-0384.1, 2017.

Clements, C. B., Lareau, N. P., Kingsmill, D. E., Bowers, C. L., Camacho, C. P., Bagley, R., and Davis, B.: The Rapid Deployments to Wildfires Experiment (RaDFIRE): Observations from the Fire Zone, Bulletin of the American Meteorological Society, 99, 2539–2559, https://doi.org/10.1175/BAMS-D-17-0230.1, 2018.

Van Heerwaarden, C. C., Van Stratum, B. J. H., Heus, T., Gibbs, J. A., Fedorovich, E., and Mellado, J. P.: MicroHH 1.0: A Computational Fluid Dynamics Code for Direct Numerical Simulation and Large-Eddy Simulation of Atmospheric Boundary Layer Flows, Geoscientific Model Development, 10, 3145–3165, https://doi.org/10.5194/GMD-10-3145-2017, 2017.